# Decline and Passive Restoration of Forest Vegetation Around the Yeocheon Industrial Complex of Southern Korea

**Hansol Lee [1], Bong Soon Lim [2], Dong Uk Kim [2], A Reum Kim [2], Jae Won Seol [2], Chi Hong Lim [3], Ji Hyun Kil [4], Jeong Sook Moon [4] and Chang Seok Lee [2,*]** 

[1] Department of Biology, Miami University, Oxford, OH 45056, USA; leeh28@miamioh.edu
[2] Department of Bio and Environmental Technology, Seoul Women's University, Seoul 01797, Korea; bs6238@swu.ac.kr (B.S.L); xellosraw@swu.ac.kr (D.U.K.); dkfma_v@swu.ac.kr (A.R.K.); seol_jaewon@swu.ac.kr (J.W.S.)
[3] National Institute of Ecology, Seocheon 33657, Korea; sync03@nie.re.kr
[4] National Institute of Environmental Research, Incheon 22689, Korea; kiljh@me.go.kr (J.H.K.); waterfa@korea.kr (J.S.M.)
[*] Correspondence: leecs@swu.ac.kr

**Abstract:** This study was carried out to clarify the vegetation decline due to air pollutants emitted in the process of industrial activities and the passive restoration of the vegetation due to socioeconomic changes after economic growth. To achieve this goal, we investigated the spatial distribution of vegetation, differences in species composition and diversity among vegetation types different in damage degree, vegetation dynamics, the age structure and annual ring growth of two dominant plant species, and the landscape change that occurred in this area over the last 50 years. Plant communities tended to be spatially distributed in the order of grassland, shrubland (dominated by *Styrax japonicus* Siebold and Zucc. community), and forests (dominated by *Pinus thunbergii* Parl. and *Pinus densiflora* Siebold and Zucc. communities), with increasing distance from the pollution source. The result of stand ordination based on vegetation data reflected the trend of such a spatial distribution. Species richness evaluated based on the species rank dominance curve was the highest in shrubland and the lowest in grassland; species richness in forests was intermediate. The size class distribution of woody plant species in four plant communities composing three vegetation types showed the possibility of them being replaced by forest in the late successional stage. However, the density of successor trees was relatively low, whereas the density of shrubby plants, which are resilient to air pollution, was very high. The age class distribution of a dominant species forming shrubland and pine forest showed that most of them were recruited after industrialization in this area. The period when young individuals in both vegetation types were recruited corresponded to the period when the annual ring growth of the pine trees that survived air pollution was reduced. An analysis of the landscape change in this area indicated that coniferous forest and agricultural field decreased greatly, whereas industrial area, residential area, mixed forest, and broadleaved forest showed increasing trends since construction of the industrial complex. As a result, the decrease in coniferous forest is usually due to vegetation decline and partially to succession, as the pine trees dominating the forest are not only sensitive to air pollution but are also shade-intolerant. The increase in mixed and broadleaved forests reflects vegetation decline or succession. Vegetation decline progressed for about 30 years after the construction of the industrial complex; it has begun to be restored passively since then, although the change has been slow. These results are in line with the environmental Kuznets curve hypothesis that environmental degradation increases in the early stages of economic growth to a certain point, and, after a turning point, economic development leads to environmental improvements—thus, there is an inverted U-shaped relationship between economic growth and environmental degradation.

**Keywords:** air pollution; industrial complex; Korea; passive restoration; vegetation damage

## 1. Introduction

Air pollution, which occurs when concentrations of substances in the air are elevated above typical background levels, causes measurable and undesirable effects on organisms and/or ecosystems [1–8]. Air pollutants come from natural sources, such as volcanic activity and forest fires, and from large-scale anthropogenic emissions. Examples of the latter include industrial processes such as metal smelting and refining, the production of commercial energy through the burning of fossil fuels, and the use of fossil-fueled engines for transportation [1,9–12]. In Korea, anthropogenic emissions of air pollutants have increased enormously over the past 50 years, due to a combination of rapid industrialization and population growth [13]. As a result, industrial and urban regions of Korea have experienced high concentrations of air pollutants, which in some cases have caused significant ecological effects, including damage to forests, like in other developed countries of the world [1,14–18].

The risks to forest ecosystems caused by air pollution are extremely complex and depend on numerous variables [1,2,7,19–23]. In cases involving exposure to high concentrations of pollutants emitted by discrete point-sources, the damage is most severe close to the point-sources, and decreases in a more or less geometric fashion with increasing distance from the source [1,2,24–26].

The study of the effects of environmental stressors on a forest ecosystem has usually focused on changes in the physiology of individuals of certain species, or, more broadly, on their population dynamics. Even less is known about the potential ecological responses to combinations of pollutants in the long term, particularly at the level of forest landscapes [27–29].

Under the stress of air pollution and acid rain, vegetation undergoes changes that may range from drastic to negligible. Their ultimate effects on vegetation structure are the results of different responses of competing plant species [1,2,4,7,16–18,26,27,30–35]. Reduced photosynthesis and visible damage such as chlorosis, lesions, and abscission are preludes to growth inhibition and to the mortality of the more sensitive plants, leading to the alteration of the vegetation structure [16–18,26,27,32,36–39]. In addition, vegetation is influenced by changes in reproductive capacity due to air pollution and acid rain [40–43], and by interactions with plant diseases and insects [44–47].

Imposed pollution stresses usually set in motion a retrogression characterized by a reduction in the structural complexity and function of vegetation [1,4,7,16–18,26,32,33,35,48–51]. Severe air pollution stress sometimes comes on so rapidly that feedback mechanisms cannot operate to select for resistant species. Nevertheless, even under severe plant competition and the imposition of air pollution stresses, some species may increase their growth, if a competitive advantage is given to them by the relatively greater impact of the pollutant on other species in the vegetation, and the growth of other species in the same vegetation may be greatly reduced because of their lowered competitive potential [1,16–18,26,30,50,52,53].

In Korea, forest ecosystems around the industrial areas under severe air pollution were degraded to shrubland, grassland, and even to denuded bare ground [16–18,33]. However, some positive signals were detected in a severely degraded forest ecosystem. For instance, grassland and shrubland turned to forest of the late successional stage correspond to the indication.

This study aims to clarify the processes of forest decline and passive restoration based on vegetation structure and dynamics that occurred in the polluted forest around the Yeocheon industrial complex in southern Korea over the last 50 years.

To arrive at this goal, first, we investigated the spatial distribution of vegetation along the distance from the industrial complex. Second, we investigated the differences of species composition and diversity among vegetation types that were different in damage degree. Third, we analyzed the dynamics of each vegetation type to predict the vegetation changes that will occur in the future. Fourth, we analyzed the annual ring growth and age structure of two major plant communities—*Pinus densiflora*

Siebold and Zucc. community and *Styrax japonicus* Siebold and Zucc. community—to clarify the background of vegetation decline. Finally, we investigated the landscape change over the last 50 years to clarify the holistic changes in the area.

## 2. Materials and Methods

### 2.1. Study Area

This study was carried out around the Yeocheon industrial complex, located on the south coasts of the Korean Peninsula (Figure 1). This study area consisted of a watershed that forms three ridges. Two ridges, which are connected from Mt. Horang to Mt. Youngchui and from Mt. Cheonseong through Mt. Buamsan to Mt. Jeseok, run in the northeast direction, and a third ridge, which stretches from Mt. Cheonseong through Mt. Bonghwa to Mt. Horang, runs in the northwest direction, connecting the other two. In the valley located between the ridges that run in the northeast direction, there are agricultural fields and residences. The elevation of the study area ranges from 0 to 510 m (Mt. Youngchui) above sea level.

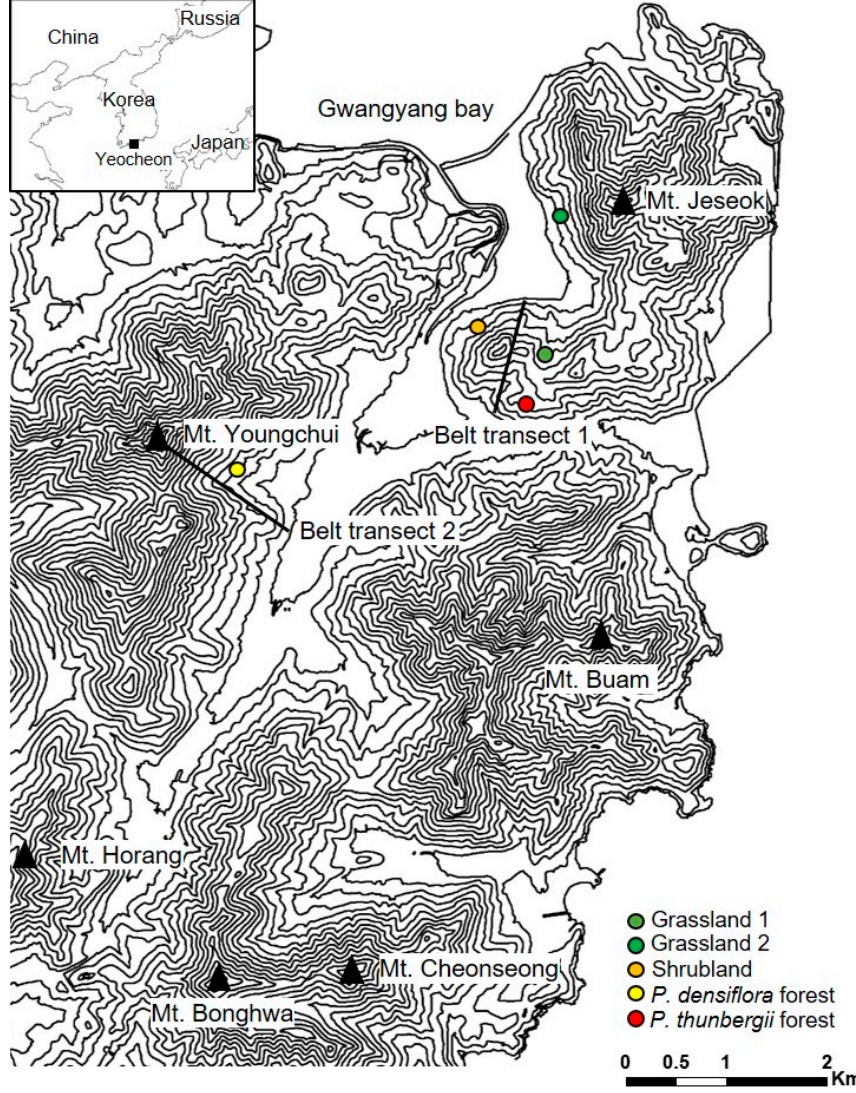

**Figure 1.** A topographic map showing the locations of belt transects and plots installed for the vegetation survey in the study area, Yeocheon industrial complex.

The parent rock of the mountainous areas consists mostly of igneous rock, and the flat land beside streams consists of alluvium. The soil in this area was classified into the alfisols, entisols, histosols, inceptisols, and ultisols, which developed on igneous bedrock [54].

The climate of this region is continental, with warm and moist summers and cold and dry winters. The mean annual temperature is 14.3 °C and the mean annual precipitation is 1428.0 mm (Korea Metrological Administration, 2020). Northeast and northwest winds dominate [55].

The forest vegetation of this area consisted of two major plant communities, the *Pinus densiflora* community in the inland area and the *P. thunbergii* community in the coastal area, before the construction of the industrial complex [56].

However, oak communities dominate areas where human interference is low and nature is thus well conserved. In this case, most areas are dominated by *Quercus mongolica* Fisch. Ex Ledeb. Community, but the *Q. serrata* Thunb. ex Murray community dominates the lower slope and the *Q. variabilis* Blum community tends to dominate the steep and infertile sites of the southern slope [57]. *Celtis sinensis* Pers. community remains as a remnant community near the coast [57].

Based on these results, we decided to use these oak communities and the *C. sinensis* community as the reference forest for comparing with the damaged vegetation types established close to the pollution source.

Yeocheon industrial complex is the representative heavy-chemical industrial complex of Korea constructed in the early 1970s. Thereafter, air pollutants that originated from many industrial facilities induced changes in the spatial distribution and structure of vegetation in the forest around the industrial complex.

Yeocheon industrial complex began construction in the early 1970s and is still expanding. Industrial activities focus on the heavy chemical industry, including the petrochemical industry. A major pollutant is $SO_2$. The $SO_2$ concentration in Yeocheon industrial complex increased continuously until 1991—the annual and daily means were 0.03 ppm and 0.06 ppm, respectively. Such severe air pollution not only caused vegetation degradation from forest to grassland and/or shrubland, but also led to soil acidification—the mean pH is 4.4 and the range is 3.5-6.7 [16,17,58]. Moreover, the soil contains lower $Ca^{2+}$ and $Mg^{2+}$ contents (one-third to one-half) and two or three times higher $Al^{3+}$ content than that in the healthy control area due to acidification [58].

However, these days, the ambient $SO_2$ concentration stays at a level below 0.01 ppm, half the standard value (annual: 0.02 ppm, daily: 0.05), the target ambient concentration set by regulation [13], and the physicochemical properties of the soil are improved [59].

On the other hand, vegetation damage of this area was also interpreted as being due to hydrofluoride [60]. In this study, the fluoride concentration in needles of the *P. thunbergii* reflected well the spatial distribution of vegetation types. Grassland, shrubland, and forest tended to be established in the sites more than 400 ppb, 200 ppb, and below 100 ppb, respectively.

## 2.2. Experimental Design

This study was carried out to clarify the processes of forest decline and passive restoration, based on vegetation structure and dynamics that occurred over the last 50 years in the polluted forest around the Yeocheon industrial complex in southern Korea. The study area was designated to be within the boundary where the visible damage of vegetation was observed. The hypothetical belt transects were installed in the first and second ridges from the pollution source to clarify the spatial distribution of vegetation and the topographic conditions (Figure 1). The plots for vegetation survey were installed in two grasslands, shrubland, and two pine forests of *P. thunbergii* and *P. densiflora*, reflecting the vegetation damage (Figure 1). We did not secure the replicated plots for each vegetation type because of location constraints. Instead, we secured the various replicated subplots for each plot. The differences of species composition and diversity were compared among vegetation types that were different in damage degree and with the reference forests. To clarify the background of vegetation decline, we analyzed the annual ring growth of *P. densiflora*, which has an easily discernible annual ring.

Furthermore, we analyzed the age structures of two dominant plants, *P. densiflora* and *Styrax japonicas*, to get information on the decline and regeneration of vegetation. Passive restoration was discussed based on the size class distribution of dominant woody plants in each vegetation type different in damage degree. The holistic changes of the area over the last 50 years were investigated by analyzing the landscape change.

### 2.3. Vegetation Survey and Data Processing

Vegetation surveys were carried out in 1991 and 2018. However, analyses of age distribution and annual ring growth, used to clarify when the forest decline started, were only performed in 1991. Field surveys were conducted from May to September in each year.

A landscape ecological map, including major vegetation types, was made based on aerial photograph interpretation (1:15,000 scale) and field checks. Field checks were done for the 1991, 2001, and 2018 maps. The final map was constructed with the ArcGIS program (ver. 10.0).

The stand profile was prepared by carefully depicting the horizontal and vertical distributions of the major plant species appearing in a hypothetical belt transect. This reflected the distance from the pollution source and changes in the topographic conditions. Spatial distribution of vegetation around the industrial area was discussed with special reference to the degree of air pollution estimated by distances from the pollution source and topographic conditions [16–18,33].

The vegetation survey was carried out by recording the cover class of plant species appearing in randomly installed quadrats of 2 × 2 m, 5 × 5 m, and 20 × 20 m in grassland, shrubland, and forest, respectively [61]. The vegetation survey was carried out in 45 subplots from June to August 1991. The numbers of subplots chosen for the vegetation survey were 9, 11, 8, 3, 6, and 8 for the *Miscanthus sinensis* var. *purpurascens* Anderss. (grassland), *Phytolacca americana* L. (grassland), *S. japonicus* (shrubland), *Celtis sinensis* Pers. (remnant forest), *Pinus thunbergii* Parl. (forest), and *P. densiflora* (forest) communities, respectively.

The 2018 survey was carried out on 48 subplots including 18, 9, 9, 5, and 7 subplots for the *M. sinensis* var. *purpurascens* (grassland 1), *S. japonicus* (shrubland), *C. sinensis* (forest), *P. thunbergii* (forest), and *P. densiflora* (forest) communities, respectively.

The reference forest, used for comparison, was designated as the oak (*Quercus serrata* Thunb. ex Murray, *Q. mongolica* Fisch. Ex Ledeb., and *Q. variabilis* Blum communities) forests, which are the representative late successional vegetation types in Korea [57]. The reference oak forests were selected in the area, which are far more than 10 km from the pollution source and therefore remain healthy. The reference forests are 50 to 100 years old, which means they are stable but not old growth. The numbers of subplots chosen for the survey for the reference oak forest were five, five, and two for the *Q. serrata*, *Q. mongolica*, and *Q. variabilis* communities, respectively.

All the plant species in each plot were identified using the Korea Plant Name Index [62]. Plant cover was recorded by applying the Braun-Blanquet [63] scale. Each ordinal cover scale was converted to the median value of percent cover range in each cover class. Relative coverage was determined by dividing the cover fraction of each species by the total cover of all species in each plot and then multiplying the value by 100. Relative coverage was regarded as the importance value of each species. A matrix of importance values for all species in all plots was constructed and used as data for ordination using a detrended correspondence analysis (DCA [64]). Plant communities were named after dominant species and followed by subdominant species in the case of a mixed community.

To compare species diversity and dominance among sites, rank abundance curves [65–67] were plotted. The data used to obtain the species rank dominance curves were collected by selecting four plots randomly from each community.

For dominant tree and shrub species, stem diameters (at breast height for mature trees, DBH; or at stem base for seedlings and saplings below 1.3 m in height, $D_0$) were measured and sorted by diameter classes. Stem diameters were measured for all tree and shrub species that appeared in four

plots (20 × 20 m) selected randomly from each vegetation type. Stem diameter was measured with a tape ruler (mature tree) or Vernier calipers (seedlings and saplings) with mm precision.

Vegetation dynamics were analyzed by a frequency distribution of diameter classes of dominant tree and shrub species. The size distributions of trees are useful indicators for understanding the structures of tree populations and for predicting their dynamics [68–70]. The diameter class distribution of plant populations has generally been computed as frequency histograms [71]. Frequency distribution patterns of each diameter class indicate the potential change of the population in a plant community. A plant population where young individuals are numerous and mature ones are fewer is recognized as having a reverse J-shaped diameter distribution pattern [72,73]. It is recognized that a population that shows a reverse J-shaped distribution pattern can persist continuously [72–75]. On the other hand, a normal population pattern with fewer juveniles relative to adults is typically replaced by another population in the future [74,75], but a bimodal pattern is shown in a population that regenerates with periodic disturbance [76,77].

Age class distribution was investigated for *P. densiflora* and *S. japonicus*, which represent vegetation types before and after construction of the industrial complex in this area, respectively. Age was determined by counting the number of annual rings of core sample collected from stem of sample plants under a dissecting microscope. Core samples were collected 30 cm above the ground surface. The time it took to reach 30 cm was determined as five and three years for *P. densiflora* and *S. japonicus*, respectively [51,78]. Age distribution was investigated for all individuals of *P. densiflora* and *S. japonicus* that appeared in four plots (20 × 20 m) selected randomly from the two vegetation types.

Core samples for estimating the tree age and analyzing the growth of annual rings were collected by an increment borer. The growth of annual rings was measured with calipers under a dissecting microscope with a 0.05 mm precision. Core samples for the analysis of radial growth were collected from 10 individuals selected randomly from the *P. densiflora* individuals that survived severe air pollution damage in the early stage when the industrial facilities began to be operated.

Landscape structure and change were interpreted by analyzing landscape ecological maps constructed before and in the 4th, 17th, 27th, and 44th years after the construction of the industrial complex.

### 2.4. Statistical Analyses

All statistical analysis was carried out using SPSS (Statistical Package for the Social Sciences) 24 software package. The *t*-test was applied for comparing the differences of the annual ring growth of *P. densiflora* between the suppressed and non-suppressed periods. Leneved's test was performed to determine the homogeneity of variance in the two groups [79].

Detrended correspondence analysis (DCA) was used to identify differences in species composition among vegetation types. DCA is a multivariate technique that maximally separates species distribution in ordination space; stand and species placements are constrained to be linear combinations of environmental variables [80]. This analysis was conducted to compare species composition among vegetation types and between the survey years 1991 and 2018.

## 3. Results

### 3.1. Vegetation Sequence

The result of a DCA ordination of stands based on vegetation data tended to be divided into the upper and the lower parts on the diagonal line that axis I and axis II form (Figure 2). Plant communities arranged in the upper part tended to be established in the upper part of the hill slope, which is far from the pollution source located near the coast, while those in the lower part were distributed in the lower part of the slope, which is closer to the pollution source and coast. In the upper part, *M. sinensis* var. *purpurascens* (grassland), *P. thunbergii* (forest), *P. densiflora* (forest), and the reference oak stands were arranged in the mentioned order in the diagonal direction. In the lower part, *P. americana* (grassland),

*S. japonicus* (shrubland), and *Celtis sinensis* Pers. (forest) stands were arranged in the mentioned order in the same direction as that in the upper part.

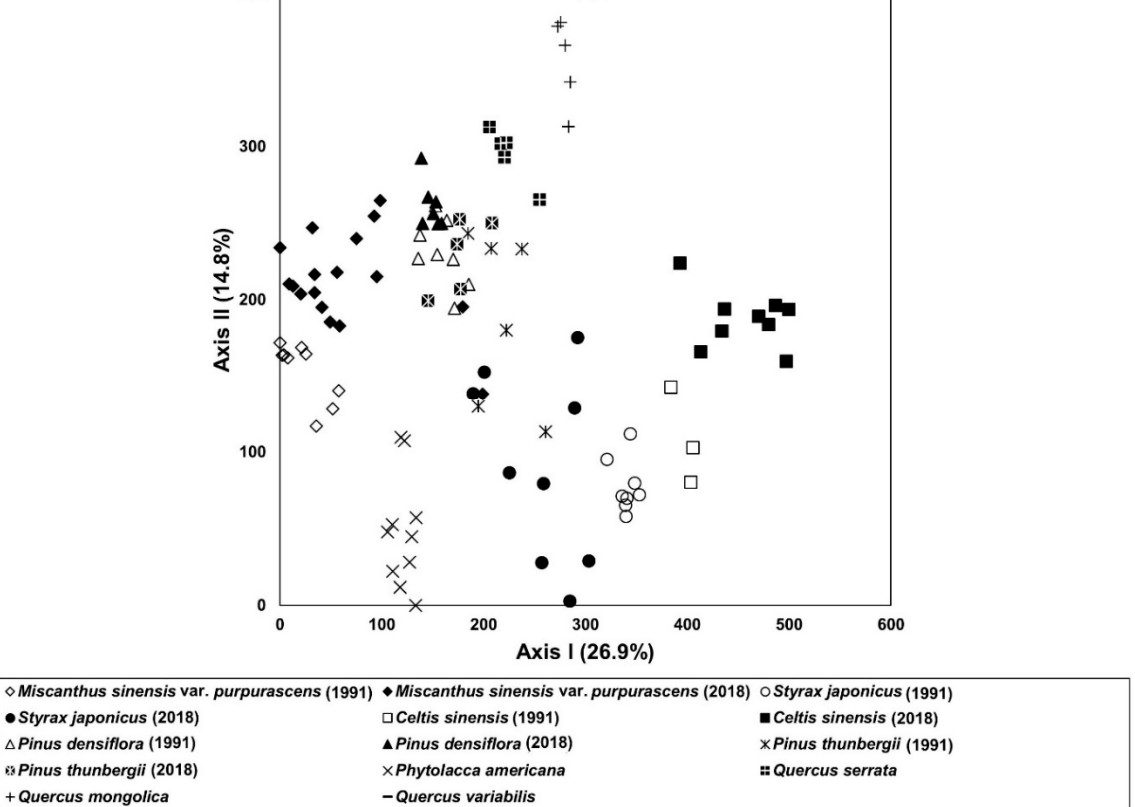

**Figure 2.** Ordination of plant communities established depending on the pollution degree around the Yeocheon industrial complex. 1991 and 2018 indicate the survey years. Vegetation types established in the inland area were arranged above the diagonal line that axis I and axis II form, whereas vegetation types established in the coastal area were below the line. On the other hand, the arrangements of vegetation types along the axis I reflect their successional stages. Synthesized those results, axis I and axis II indicate the degree of pollution and vegetation degradation due to the distance from the pollution source and the distance from the coastal area, respectively.

In the result based on data collected in 2018, stands of *M. sinensis* var. *purpurascens* tended to be closer to the forest stands composed of *P. thunbergii* and *P. densiflora*, whereas stands of *P. thunbergii* and *P. densiflora* little changed. Stands of *S. japonicus* became far from the forest stands, which were composed of *C. sinensis* and scattered. Stands of *C. sinensis* became farther from stands of *S. japonicus*; stands succeeded from *P. americana* stands were arranged close to stands surveyed in 1991 and *S. japonicus* stands. On the other hand, stands of *P. americana* were succeeded by *C. sinensis*.

### 3.2. Species Diversity

In the data from 1991, species richness was the lowest in the *M. sinensis* var. *purpurascens* community and the highest in the *S. japonicus* and *P. thunbergii* communities; the *P. americana* and *P. densiflora* communities were at a medium level. Compared with the reference oak communities, the species richness of *M. sinensis* var. *purpurascens* and *P. densiflora* communities was the lower but those of *S. japonicus* and *P. thunbergii* communities were similar to each other (see the x-axis in Figure 3).

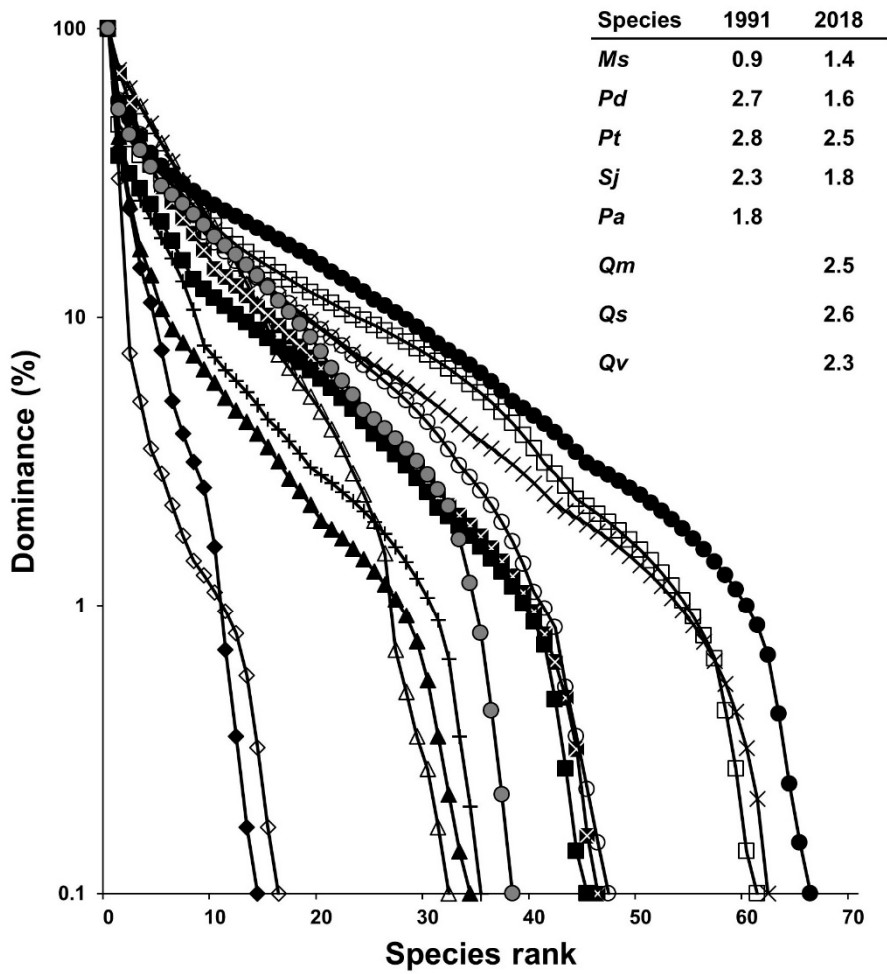

| Species | 1991 | 2018 |
|---|---|---|
| *Ms* | 0.9 | 1.4 |
| *Pd* | 2.7 | 1.6 |
| *Pt* | 2.8 | 2.5 |
| *Sj* | 2.3 | 1.8 |
| *Pa* | 1.8 | |
| *Qm* | | 2.5 |
| *Qs* | | 2.6 |
| *Qv* | | 2.3 |

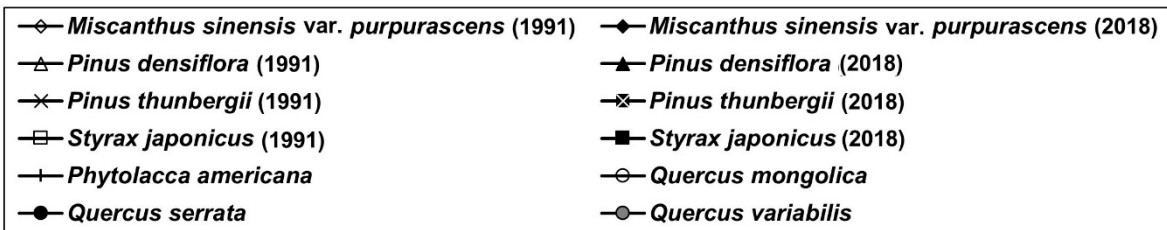

**Figure 3.** Species rank dominance curves of plant communities established around the Yeocheon industrial complex. 1991 and 2018 indicate the survey years. The numbers in the inset indicate the Shannon indices. Ms: *Miscanthus sinensis* var. *purpurascens* stand, Pd: *Pinus densiflora* stand, Pt: *P. thunbergii* stand, Sj: *Styrax japonicus* stand, Pa: *P. americana* stand, Qm: *Quercus mongolica* stand, Qs: *Q. serrata* stand, Qv: *Quercus variabilis* stand.

In the data from 2018, the order of species richness did not change. Comparing the species richness between survey years, those of the *S. japonicus* and *P. thunbergii* communities decreased greatly but those of the *M. sinensis* var. *purpurascens* and *P. densiflora* communities changed little.

The results compared by the slope of the species rank dominance curve, which indicates the evenness, showed a similar trend (Figure 3).

The results compared based on Shannon index showed a similar trend (Figure 3).

### 3.3. Vegetation Dynamics

In the 1991 survey for grassland 1, oaks, which are usually *Quercus serrata*, appeared in a state of undergrowth below dominant *M. sinensis* var. *purpurascens*, and the density was also very low.

The size class distribution of them showed a reverse J-shaped pattern. In the 2018 survey, *Quercus* spp. grew to shrub size but not many new seedlings were recruited; therefore, the size class distribution showed a J-shaped pattern (Figure 4).

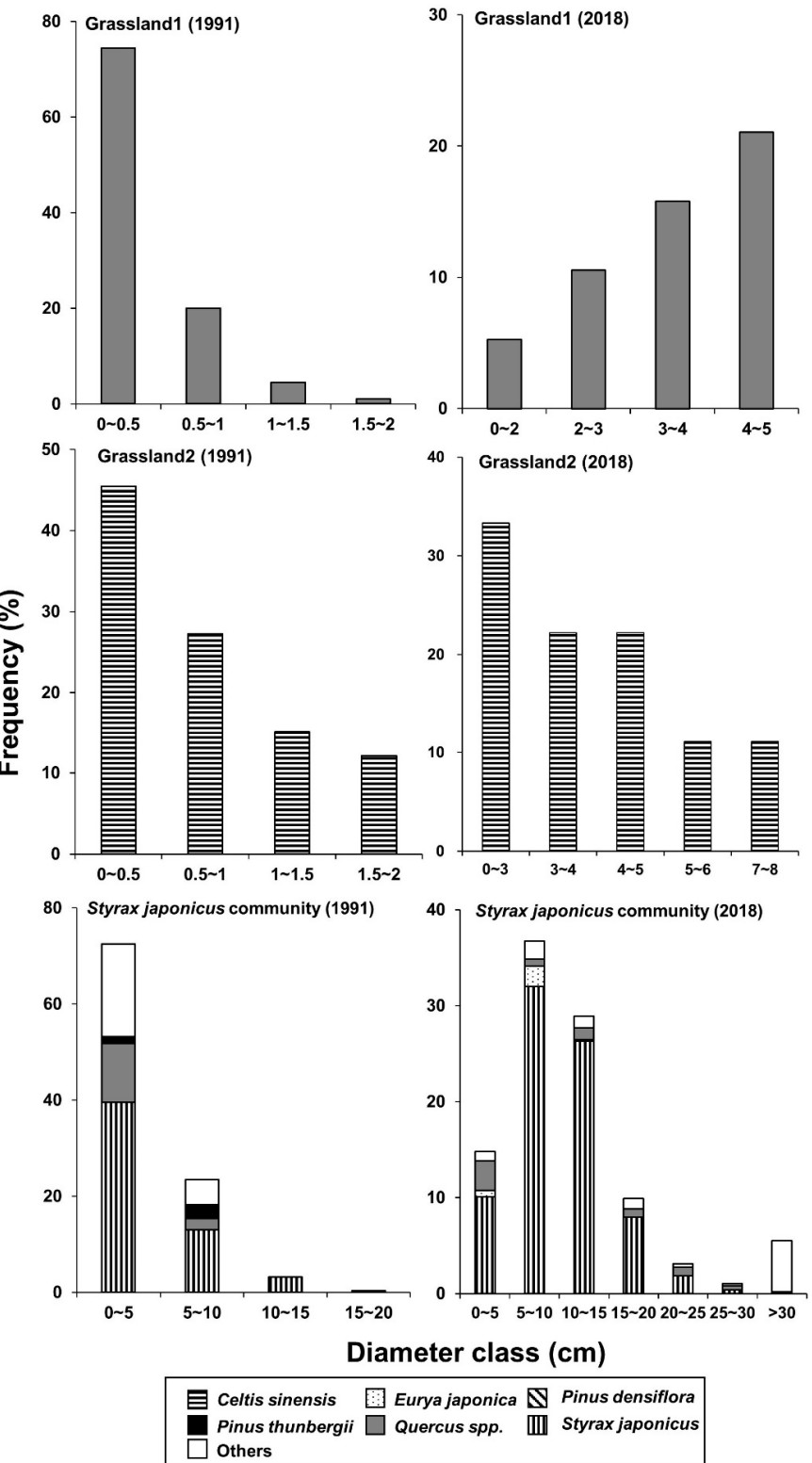

**Figure 4.** *Cont.*

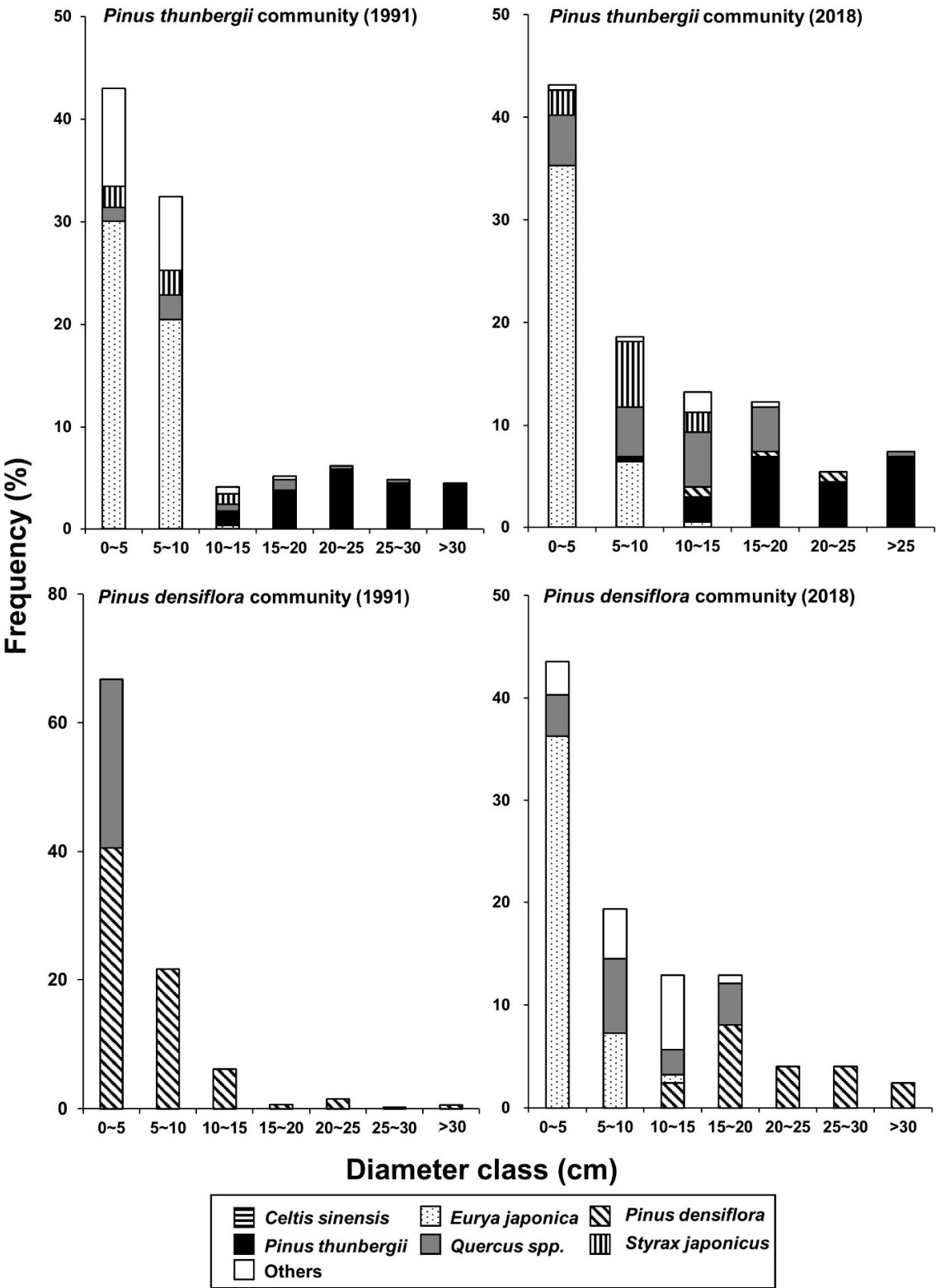

**Figure 4.** Frequency distribution diagrams of diameter class of tree and two shrub species in three vegetation types: grassland 1 (*M. sinensis* var. *purpurascens* community), grassland 2 (*P. americana* community), shrubland (*S. japonicus* community), and forest (*P. densiflora* community, *P. thunbergii* community).

In grassland 2, which is located near the northern coast, *C. sinensis* appeared. *C. sinensis* grew bigger than oaks and their size class distribution showed a reverse J-shaped pattern in which new seedlings were recruited continuously in the 2018 survey (Figure 4).

In the size class distribution diagram for the dominant species composing *S. japonicus* stands, *S. japonicus* showed a reverse J-shaped distribution pattern. *Quercus* spp. had a frequency of over 10% in diameter classes below 10 cm, but its frequency in other classes was low. In this diagram, *Pinus* spp. also appeared, but their frequency was also low. In the 2018 survey, the *S. japonicus* population still showed a reverse J-shaped pattern, except in the diameter class below 5 cm. *Quercus* spp. appeared in all diameter classes evenly; mature trees more than 30 cm also appeared, although the frequency was not so high (Figure 4).

In the size class distribution diagram for dominant species composing a *P. thunbergii* forest located on the opposite slope of the pollution source, the *P. thunbergii* population showed the normal distribution pattern, with fewer juveniles relative to adults, whereas the *Eurya japonica* Thunb. population showed a reverse J-shaped distribution pattern in diameter classes below 15 cm. *Quercus* spp. usually appeared in the lower diameter classes, but the density was very low. *S. japonicus* was restricted to diameter classes below 15 cm and showed a frequency distribution similar to that of *Quercus* spp. In the 2018 survey, the size class distribution diagram changed little, but the oak density increased a bit (Figure 4).

In the size class distribution diagram for dominant species composing *P. densiflora* forest, the *P. densiflora* population showed a reverse J-shaped pattern whereby seedlings and saplings are recruited continuously. *Quercus* spp. appeared only in the diameter classes below 5 cm. In the 2018 survey, the *P. densiflora* population changed to a normal distribution pattern, with fewer juveniles than adults. The *E. japonica* population as an evergreen shrub appeared at a very high density below 10 cm and showed a reverse J-shaped pattern. *Quercus* spp. showed a reverse J-shaped pattern, although the density was relatively low (Figure 4).

### 3.4. Yearly Changes in Annual Rings

The annual ring growth of pine trees that survived air pollution damage was suppressed for about 10 years after industrial facilities began to operate in this area, but since then they tended to recover (Figure 5). It was supposed that the suppressed growth originated from air pollution and the improvement of growth since then was due to the release from competition after the selective death of neighboring trees and the mitigation of air pollution [51,73].

Annual ring growth during the suppressed period showed a significant difference from that of the non-suppressed period (Table 1).

**Table 1.** The result of *t*-test for the difference in annual ring growth of *P. densiflora* between the suppressed and the non-suppressed periods.

| Period | Mean (mm) | Standard Deviation (mm) | *t*-Value | *p* |
|---|---|---|---|---|
| Suppressed (1974 to 1985) | 2.64 | 1.49 | 6.99 | 0.00 |
| Non-suppressed (other period) | 4.12 | 1.71 | | |

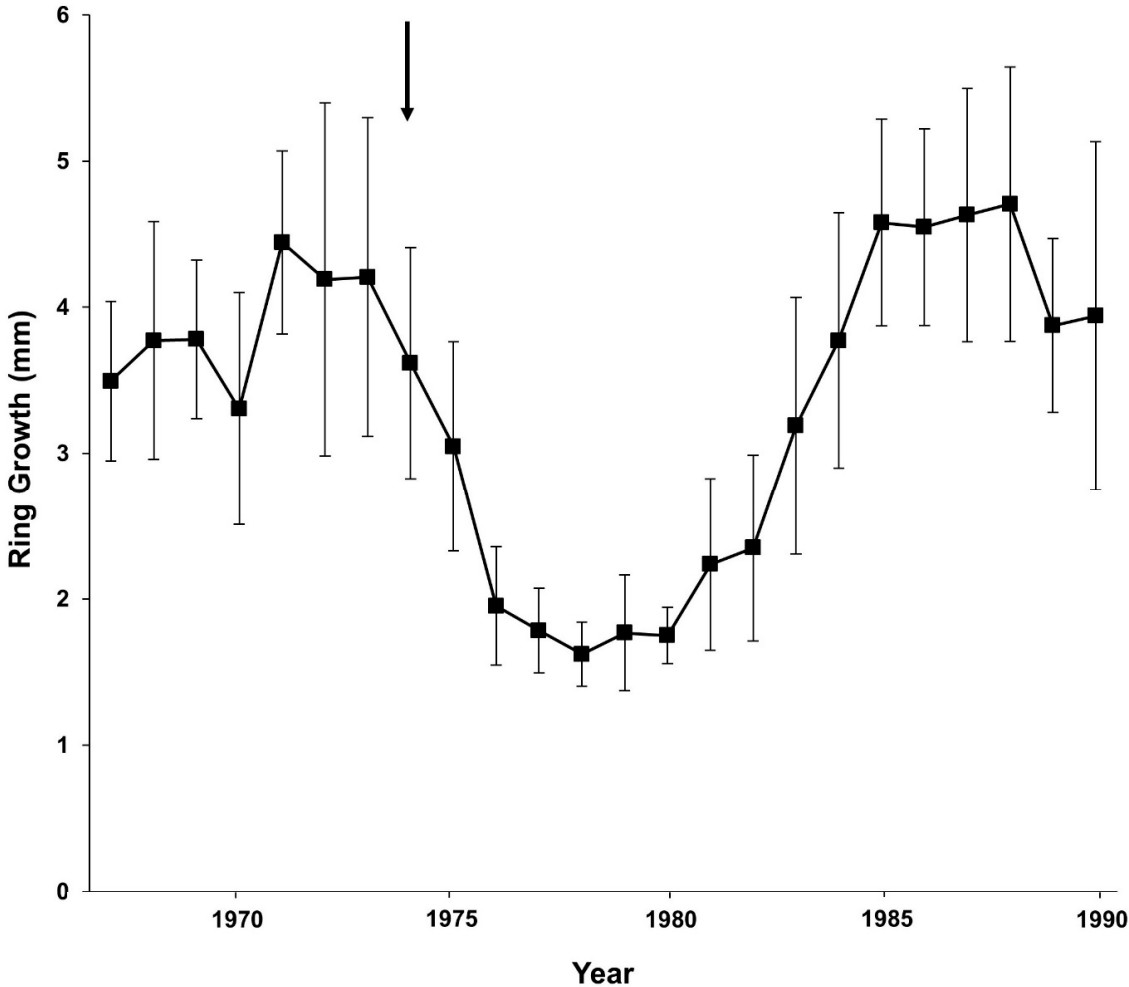

**Figure 5.** Yearly changes in radial growth of *P. densiflora* sampled around the Yeocheon industrial complex. The arrow indicates when the industrial facility began to operate in this area.

*3.5. Age Distribution of Two Dominant Species*

The age of *S. japonicus* ranged from five to 37 years (Figure 6). Individuals younger than 18 years, established after the construction of the industrial complex, made up more than 70% of the total.

In an age class distribution diagram of the *P. densiflora* community (Figure 6), the ages of the pine trees ranged from one to 33 years. Those aged 10 to 15 years old accounted for more than 40%.

The period when young individuals were recruited in both the *S. japonicus* and *p. densiflora* communities corresponded to the period when the annual ring growth of the pine trees that survived air pollution was suppressed (Figure 6). These results suggest that most individuals in these two communities are the products of air pollution damage and natural regeneration of damaged forest.

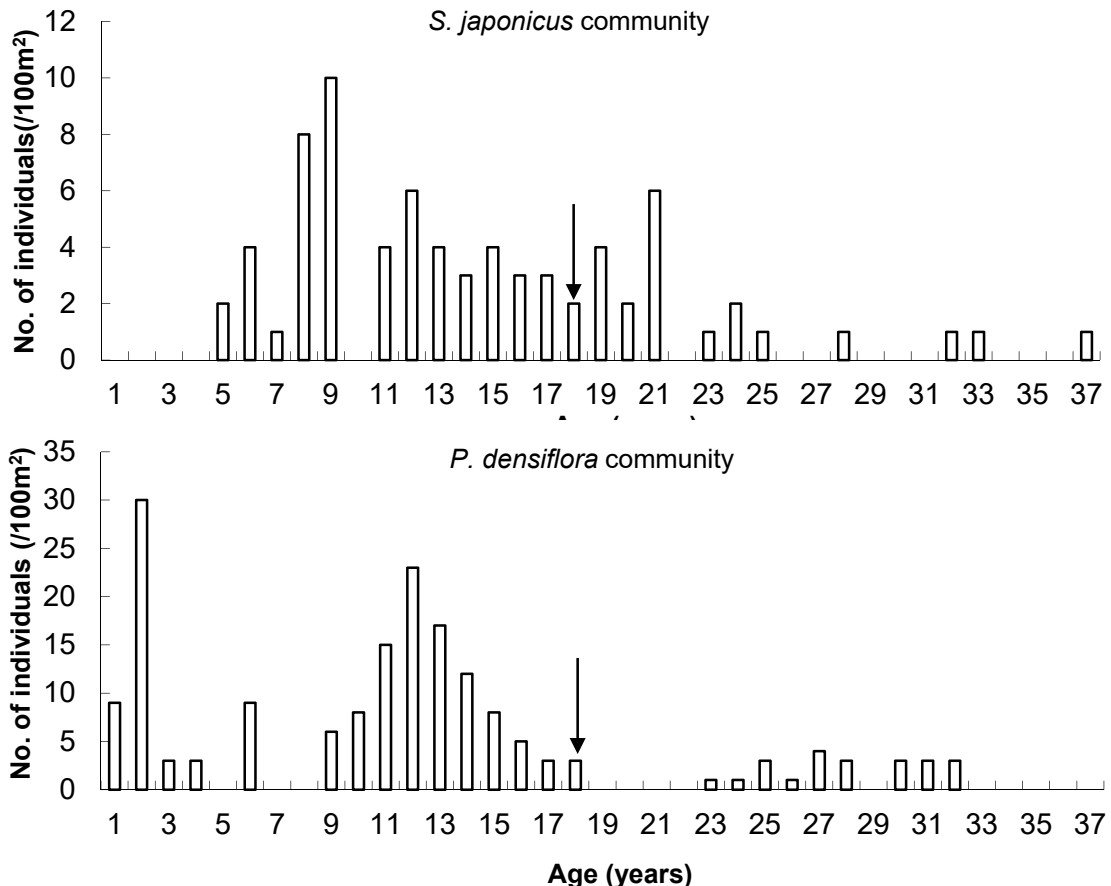

**Figure 6.** Age distribution diagrams of two woody plant communities, *S. japonica* community and *P. densiflora* communities, established around the Yeocheon industrial complex. Arrow indicates the year that the industrial facility in this area began to operate in this area.

### 3.6. Landscape Structure and Change

The land use pattern in this area is divided clearly depending on the topographic conditions. Upland was covered with forest, whereas lowland was dominated by agricultural field until the early 1970s. However, since then, an industrial area appeared as a new landscape element. The industrial area was restricted to the northern lowland in the early 1970s, but has expanded continuously since then.

Forest in this area before the construction of industrial complex was usually coniferous, dominated by the Korean red pine (*P. densiflora*) community and partially mixed with black pine (*P. thunbergii*) depending on the site [56]. Grassland began to appear near the northern part of the industrial complex in the late 1970s. The area occupied by grassland increased continuously throughout the 1990s and early 2000s but decreased in recent years. Grassland appeared on the ridge distant from the industrial area as well. Shrubland appeared in 1991; its area increased up to the early 2000s but decreased in recent years, as was the case with grassland. In 2018, the change from coniferous forest to mixed or deciduous forest was remarkable. In addition, the plantation and residential area also increased (Figure 7).

In 1971, before the construction of the industrial complex, coniferous forest dominated by *P. densiflora* occupied the largest area at 72.3%, and agricultural fields (23.6%), residential areas (2.2%), broad-leaved forests (1.4%), and so on followed.

By 1978, the landscape structure was little changed, except that industrial areas (3.5%) and grassland (1.6%) were new categories.

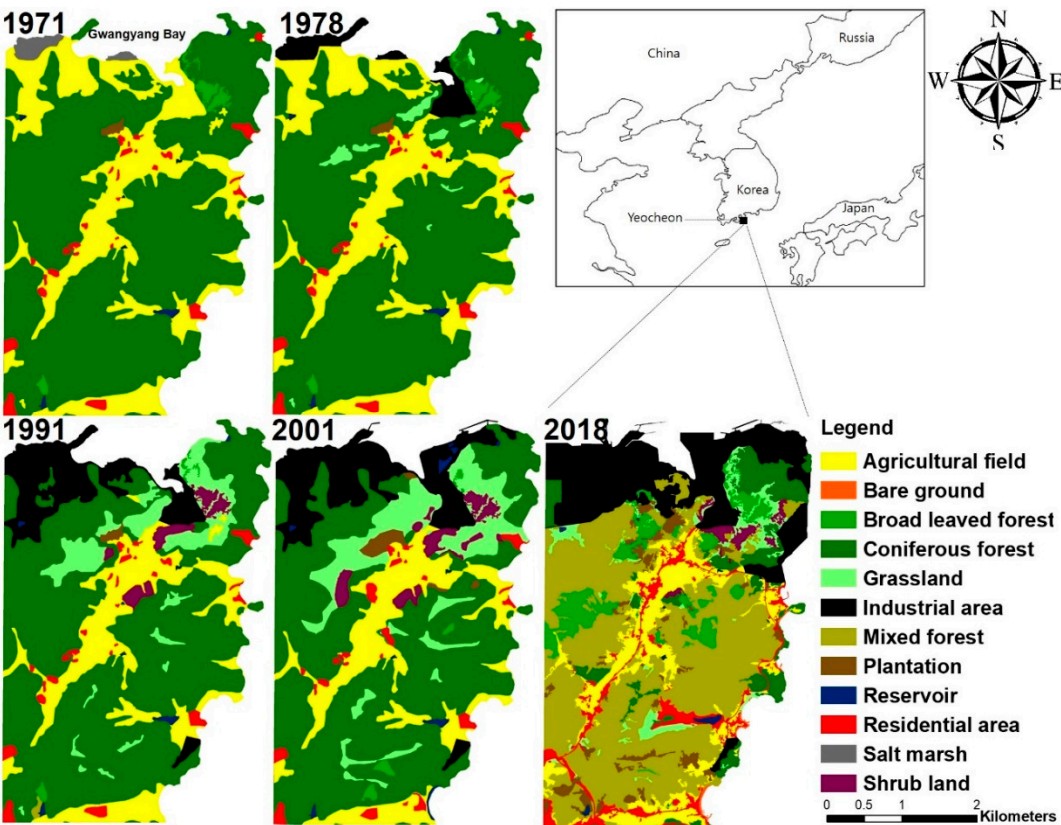

**Figure 7.** Maps showing the landscape change near the Yeocheon industrial complex.

In 1991, the 17th year after the industrial facility was constructed, the areas occupied by coniferous forest and agricultural fields were reduced to 55.4% and 14.9%, respectively, whereas those of grassland and industrial increased greatly to 14.7% and 9.4%, respectively. On the other hand, shrubland dominated by the *S. japoncus* community (1.8%) appeared for the first time in 1991.

In 2001, coniferous forest decreased to 53.1%, whereas the industrial area increased to 11.9%. The area of shrubland (2.6%) also increased, although not so noticeably.

In 2018, the areas of coniferous forest and grassland decreased greatly to 8.1% and 3.8%, respectively; mixed forest (42.2%) increased greatly, and broad-leaved forest (6.3%) and plantation (3.3%) also increased. The area of agricultural field decreased more, whereas that of industrial area increased more. The area of shrubland (1.3%) decreased a little. In this period, the increase in mixed forest was due to the increase in broad-leaved forest composed of oaks and *C. sinensis*. The former was usually established above the mid-slope, while the latter was in lowland close to the northern coast.

The landscape elements identified in each investigated year and landscape change are summarized in Appendix A.

## 4. Discussion

### 4.1. Characteristics and Establishment Background of Vegetation Around the Yeocheon Industrial Complex

In a heavily industrialized area, the effects of high levels of atmospheric pollution may be demonstrated quite visibly in surrounding forests and other ecosystems [1,2,81–84]. In such cases of intense air pollution, vegetation may suffer extensive mortality, resulting in dramatic changes in species composition, reductions in biomass and productivity, erosion of soil, nutrient loss, and changes in microclimatic and hydrological regimes. In severe cases, the perturbation can result in extreme ecosystem simplification and almost complete denudation [1,2,4,26,85].

Grassland, dominated by *M. sinensis* var. *purpurascens*, *P. americana*, etc., and shrubland, dominated by *S. japonicus*, which emerged since the construction of the industrial complex in this study area, reflect such phenomena (Figure 7). A structural change from forest to shrubland or grassland means a reduction in biomass (Figure 7) and vegetation simplification (Figure 4).

*M. sinensis* var. *purpurascens* is extremely shade-intolerant, so it cannot form a pure stand throughout this wide area if there is no bare ground. *P. americana* is an exotic plant. It is rare and also cannot form a pure stand of wide area if there is no severe disturbance. However, the density increased closer to the pollution source and it formed pure stands in sites near the pollution source. *S. japonicus* has similar ecological attributes to those plant species. NIER [86] classified *S. japonicus* as a species sensitive to pollution, whereas NIER [87] did it as a tolerant species. This difference might be related to the period in which the observation is conducted. As shown in our results (Figures 6 and 7, Table 1), this species appeared since the construction of the industrial complex. This means that this species is tolerant of the environmental conditions in this area. In an experimental study of exposure to $SO_2$, *S. japonicus* was sensitive to $SO_2$ exposure, showing rapid necrosis and leaf shedding [58]. However, the damaged plants produced new leaves a few days later. This implies that this plant is a resilient to air pollution rather than a tolerant species [88]. *C. sinensis*, which emerged in grassland 2, also showed the characteristics of a resilient species.

Direct damage to vegetation near point sources is characterized by a spatial pattern of exponentially decreasing intensity with increasing radial distance. Acute effects on vegetation may occur in areas immediately surrounding sources of emitted pollutants, while lower-level, regional effects may be much more widespread, but difficult to document. The direct and indirect effects of air pollutants on forest ecosystems depend on numerous complex variables, including the intensity of exposure; mitigating environmental conditions; and the inherent susceptibility of affected populations, species, and communities [7,89,90]. The spatial distribution of vegetation in this area, which was dominated by the distance from the pollution source and topography, reflects these trends (Figures 7 and 8).

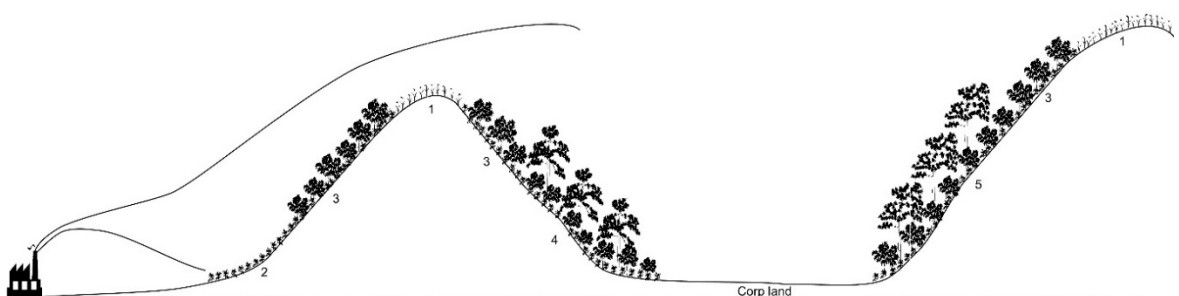

**Figure 8.** A diagram showing the spatial distribution of vegetation along the distance from pollution source and the topographic conditions in Yeocheon industrial complex. 1: *M. sinensis* var. *purpurascens* community; 2: *P. americana* community; 3: *S. japonicus* community; 4: *P. thunbergii* community; 5: *P. densiflora* community.

The spatial distribution of vegetation along the distance from the pollution source tended to be in the order of grassland, shrubland (dominated by *S. japonicus* community), and forest (dominated by *P. thunbergii* and *P. densiflora* communities) (Figure 8). Such a distributional trend might be related to the air pollutants being transported with the land and sea breeze as it is a coastal area bordering Gwangyang bay (Figures 1 and 7) [17]. Grassland was dominated by the *Miscanthus sinensis* var. *purpurascens* and *Phytolacca americana* communities. This *M. sinensis* var. *purpurascens* community is a vegetation type that emerged from damage due to severe air pollution. Shrubland was dominated by the *S. japonicus* community. This community is established in areas near to the pollution source, but the effects of air pollution are mitigated by topographic conditions. Forests, dominated by *P. thunbergii* and *P. densiflora* communities, were established in the areas which are distant from the pollution source or

experience more mild air pollution due to the topographic conditions, such as at the foot of the slope opposite the pollution source (Figure 8).

This phenomenon was also reflected on the result of ordination based on the vegetation data (Figure 2). In the result of stand ordination, vegetation types established in the inland area were arranged above the diagonal line that axis I and axis II form, whereas vegetation types established in the coastal area were below the line. The vegetation types changed from grassland through two pine forests to the late successional oak forests above the line, and from grassland through shrubland and some pine forests to the *C. sinensis* forest in the lower part as move along the diagonal direction on the plane that axis I and axis II form. From these results, we could interpret that axis I means the successional stage and axis II reflects the distance from the coastal area (Figure 2). That is, the distribution and composition of vegetation in this area are usually dominated by the distance from the source of pollution, and the influence of the sea is partial. In particular, the fact that the result of ordination reflects the successional stage means that pollution affects not only the distribution of vegetation but also the progression of the succession. Previous studies carried out in this area revealed that soil acidification, changes of physicochemical properties of soil due to that, and visible damage of plants are usually dominated by the distances from the pollution source [17,18,32,37]. Considering the facts synthetically, axis I indicates the degree of pollution and vegetation degradation due to the distance from the pollution source.

The spatial distribution of vegetation and the difference of species composition among vegetation types do not have a statistical significance because we did not secure the replicated belt transects and plots for each vegetation type due to location constrains. However, we can read those trends from the spatial distribution diagram (Figure 8), the result of ordination (Figure 2), and the vegetation map (Figure 7).

### 4.2. Relationship Between Environmental Factors and Spatial Distribution of Vegetation

In most studies on point sources of air pollution, there is a generic observation that ecological damage becomes progressively less severe as the distance from the source increases. This pattern closely tracks the gradients of pollution, which are characterized by a roughly exponential decrease in intensity as the distance from the source increases [1].

If a forested ecosystem is affected by air pollution, the canopy level is generally impacted first and is stripped away. As the canopy declines, shrubs and then the ground vegetation are affected. This syndrome of the sequential death of horizontal strata of the terrestrial vegetation, described as a "peeling off" or "layered vegetation effect" by Gordon and Gorham [91], was clearly observed in this area (Figure 8).

Under long-term pollution stress, the ecosystem of this area was characterized by biological simplification. In addition, pollution-tolerant species are not usually present or are rare, since unpolluted habitat dominates the vegetation of this area. Most of them began to emerge as industrial facilities were opened in this area (Figures 6 and 7). Forest remnants remained in sites where the topography gave some protection from fumigation. They were on relatively protected, mesic, lower slopes of hills (Figure 8).

Grassland and shrubland emerged as a result of severe air pollution damage (Figures 5–7), and forest remnants that survived the damage were impoverished around the industrial complex (Figures 3 and 8). A sharp decrease in species richness was observed in grassland, and richness also decreased in *P. densiflora* forest (Figure 3). The decrease in grassland was due to the simplification of the vegetation structure from "layered vegetation effect" [91]. The high density of *E. japonica* indicated that it was tolerant to air pollution [16,17,33,58]. On the other hand, the species richness of the *S. japonicus* and *P. thunbergii* communities was high—at a similar level as the reference oak communities (Figure 3). This result can be explained based on the intermediate disturbance hypothesis that species diversity is maximized when ecological disturbance is neither too rare nor too frequent [92].

*4.3. Future Prospects Based on Vegetation Dynamics*

In grassland 1, diameter class distribution of *Quercus* spp. in grassland showed a reverse J-shaped pattern, although the density was very low (ca. 200 individuals·ha$^{-1}$). In the 2018 survey, the density increased to 350 individuals·ha$^{-1}$ and the diameter also increased to the level of the shrub layer, but new seedlings were not recruited often, and therefore the size class distribution showed a J-shaped pattern (Figure 4).

In grassland 2, which is located near the northern coast, *C. sinensis* appeared as a successor tree in the 1991 survey. *C. sinensis* grew a little bigger than oaks and the size class distribution showed a reverse J-shaped pattern in the 2018 survey (Figure 4).

*S. japonicus*, a dominant species of the *S. japonicus* community, showed a reverse J-shaped pattern in the 1991 survey. *Quercus* spp. also showed a reverse J-shaped pattern in this stand, but their density was very low compared with *S. japonicus*. In the 2018 survey, recruitment of *S. japonicus* seedlings decreased but still showed a reverse J-shaped pattern. *Quercus* spp. appeared evenly in all diameter classes and mature trees over 30 cm also appeared, although the frequency was still low. *Pinus* spp. appeared in 1991 but disappeared in 2018.

In the *P. thunbergii* forest, the frequency distribution of the diameter class of the *P. thunbergii* population showed a normal pattern without seedling recruitment in both the 1991 and 2018 surveys. The density of *Quercus* spp. as successor trees was very low in the 1991 survey but increased in 2018. However, shrubby plants, such as *Eurya japonica* and *S. japonicus*, which are tolerant or resilient to air pollution, dominated the smaller diameter classes.

In the *P. densiflora* forest, the *P. densiflora* population showed a reverse J-shaped pattern in 1991 but changed to a normal distribution in 2018. On the other hand, *Quercus* spp., which appeared only in the diameter class below 5 cm in 1991, showed a reverse J-shaped pattern, although their density was relatively low in 2018. Even in this forest, two shrubby plants dominated the smaller diameter classes.

From those results, we could deduce that oaks or *C. sinensis* would be the potential dominant species in this region (Figure 4). In grassland, both the density and size of oaks increased in 2018 compared with 1991, although the density was still very low. Meanwhile, the *C. sinensis* emerged as a successor tree in grassland located near the coast. Even in the results of *S. japonicus* stands collected in 2018, the density of oaks increased, and some individuals grew to the canopy level. In forests dominated by the *P. thunbergii* and *P. densiflora* communities, oaks showed potential to be successor trees, but their density was relatively low. Moreover, shrubby plants dominated smaller diameter classes with very high density. Therefore, whether the vegetation of this area, which is still affected by air pollution from the industrial complex, could be succeeded by oak forest in the late successional stage along the normal successional trajectory or not depends on the intensity of pollution; the environmental quality in this area is getting better [13,78]. Before industrialization, this area was covered in mixed pine forests [56]. Such pine domination was due to frequent human disturbances typical of rural areas in Korea—for example, cutting, pruning, weeding, needle scraping to get timber, fuel, organic fertilizer, feed for livestock, etc. [93,94]. However, socioeconomic changes due to the rapid economic growth since the 1980s vanished these land uses in Korea. Consequently, many coniferous forests are in succession to broad-leaved forest, including oak forest, these days; a similar successional trend was confirmed in this industrial area as well.

*4.4. Implications for Restoration*

Air pollutants emitted in the processes of industrial activities simplify the vegetation structure through layered vegetation effects, which parallel the pollution intensity [91,95] and further weaken the function [1,2,4,16–18,33,50]. Vegetation fulfills diverse ecological functions, including air purification [7,96,97]. Therefore, the structural simplification and functional weakening of vegetation aggravates pollution damage [1,7,16–18,33,50]. From this viewpoint, the recovery of ecosystem services through vegetation restoration is a necessity.

Restoration is the process of assisting in the recovery and management of ecological integrity [98,99]. A restoration project can range from a very simple task to a very complex science. The methods are divided as follows depending on the degree of human intervention [100]. The first method relies on the natural recovery ability of the ecosystem. The second one facilitates vegetation recovery, at least by investing bioenergy. The last one is restoring the ecosystem by investing bioenergy positively, such as through the planting of seedlings.

In order to restore a degraded ecosystem in a polluted environment, interception of pollutants, the amelioration of the polluted environment, the selection of pollution-tolerant plants, the determination of the best planting methods, etc., are required [16–18,33,50,101]. The structure of the plant community is determined in response to the abiotic environment that is established. Therefore, clarifying the structural characteristics of the plant communities that are established in the polluted environment could provide crucial information that can be used to determine the methods and kinds of plants to be introduced to restore the vegetation [98].

As mentioned above, woody plants, which have the potential to be dominant in this region, appear on the vegetation floor in grassland. Even in shrubland or pine-dominated forest where there are better environmental conditions than in grassland, they keep their status as potentially dominant. This implies that natural vegetation can be recovered through passive restoration [102,103]. However, vegetation change is far slower than during active restoration [18] or after the succession of pine forest, which involved pine gall midge damage [51]. Such a situation suggests that restoration, the second of the three methods that Bradshaw [100] introduced, would be suitable for this area. When restoration was practiced by introducing potential dominants, which were selected as tolerant species through several experimental procedures [16,17], it showed restoration effects in terms of a change in species composition and an increase in biodiversity [18].

## 5. Conclusions

We reached to the following conclusion from the analyses on the spatial distribution of vegetation, differences in species composition and diversity among vegetation types different in damage degree, vegetation dynamics, the age structure and annual ring growth of two dominant plant species, and the landscape change that occurred in this area over the last 50 years.

The distribution and composition of vegetation are affected by the distance from a source of pollution. Our results also reflected the successional stage of the plant communities, which means that pollution affects not only the distribution of vegetation, but also the progression of the succession.

The results of analyses on the annual ring growth and the age structure of two dominant species showed that the establishment of shrubland and forest—composed of shade-intolerant plants, such as pines—is a result of vegetation decline due to pollutants discharged after the construction of the industrial complex, and regeneration of the gap occurred from the vegetation's decline.

The size class distribution of dominant woody plants indicated that it was likely that all vegetation types, including grassland, shrubland, and pine forest, would be succeeded by deciduous broadleaved forests in the late successional stage as the effects of pollution diminish. These results can be interpreted as the result of the passive restoration that occurred as the environmental conditions improved due to socioeconomic changes. Furthermore, this trend is also confirmed by the analysis of landscape changes.

**Author Contributions:** Writing—original draft, H.L.; Investigation, Data curation and Formal analysis, B.S.L., D.U.K., A.R.K., J.W.S., J.S.M., J.H.K.; methodology and software, C.H.L.; Conceptualization, Supervision, Writing—review & editing, C.S.L.; All authors have read and agreed to the published version of the manuscript.

**Funding:** This research received no external funding.

**Conflicts of Interest:** There are no potential conflicts of interest to declare.

## Appendix A  Landscape Change Progressed for the Last 50 Years around the Yeocheon Industrial Complex

| Landscape Element | 1971 | | 1978 | | 1991 | | 2001 | | 2018 | |
|---|---|---|---|---|---|---|---|---|---|---|
| | km$^2$ | % | km$^2$ | % | km$^2$ | % | km$^2$ | % | km$^2$ | % |
| Forest | | | | | | | | | | |
| Broad leaved forest | 0.6 | 1.1 | 0.6 | 1.1 | 0.2 | 0.3 | 0.3 | 0.6 | 3.6 | 6.3 |
| Coniferous forest | 38.5 | 71.4 | 37.8 | 69.9 | 33.2 | 61.0 | 30.4 | 53.7 | 4.6 | 8.1 |
| Mixed forest | - | - | - | - | 0.5 | 0.9 | - | - | 24.0 | 42.2 |
| Plantation | 0.1 | 0.3 | - | - | 0.1 | 0.3 | 0.6 | 1.0 | 1.9 | 3.3 |
| Shrub land | - | - | - | - | 1.1 | 2.0 | 1.5 | 2.7 | 0.7 | 1.3 |
| Grassland | - | - | 0.8 | 1.5 | 4.3 | 7.9 | 6.4 | 11.3 | 2.2 | 3.8 |
| Subtotal | 39.2 | 72.7 | 39.2 | 72.5 | 39.3 | 72.3 | 39.2 | 69.3 | 36.9 | 65.0 |
| Others | | | | | | | | | | |
| Agricultural field | 12.7 | 23.6 | 11.6 | 21.4 | 8.5 | 15.5 | 8.5 | 15.0 | 6.1 | 10.8 |
| Bare ground | - | | - | | - | | - | | 0.1 | 0.2 |
| Industrial area | - | | 1.9 | 3.6 | 5.3 | 9.8 | 6.8 | 12.1 | 10.4 | 18.4 |
| Reservoir | 0.2 | 0.4 | 0.2 | 0.4 | 0.2 | 0.4 | 0.4 | 0.8 | 0.2 | 0.3 |
| Residential area | 1.2 | 2.2 | 1.2 | 2.2 | 1.1 | 2.1 | 1.6 | 2.9 | 3.0 | 5.3 |
| Salt marsh | 0.6 | 1.2 | - | | - | | - | | - | |
| Subtotal | 14.7 | 27.3 | 14.9 | 27.5 | 15.1 | 27.7 | 17.3 | 30.7 | 19.9 | 35.0 |
| Total | 54.0 | 100.0 | 54.1 | 100.0 | 54.5 | 100.0 | 56.5 | 100.0 | 56.8 | 100.0 |

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
