# Peer review of "Decline and Passive Restoration of Forest Vegetation Around the Yeocheon Industrial Complex of Southern Korea"

_forests, doi:10.3390/f11060674_

Round 1

Reviewer 1 Report

Forests Review: Manuscript ID: forests-795091 

Overall comments: The paper reports on shifts in vegetation over time as a result of air pollution sourced in industrial activity in the Yeocheon industrial complex in Korea. The research presents interesting and important findings and is certainly appropriate for Forests, with major revision recommended. The paper could benefit from some editing for grammar and usage in some places—syntax is at times awkward.  

Specific comments:  

Abstract:  

  • The abstract is overly focused on the important results of the study, and lacks a treatment of the context and methods of the study. I suggest rewriting the abstract as a brief summary of the entire paper, including all major sections.  

Introduction:  

  • Pg 2, paragraph 3: “The study... have usually focused” (example of English usage problem: the subject and verb should agree in number 
  • The authors describe some of the specific air pollution types and effects in the Materials and Methods section, but I suggest that some of this may be more appropriately relocated to the Introduction—for example, much of the paragraph on SO2 emission and its effect on soils. Also, is SO2 the only pollutant of concern in this region? This is the only specific pollutant mentioned.  

Methods and Materials:  

  • Figure 1 seems to be also the results of the analysis of landscape change, which is listed as the final objective in the introduction. This figure may be better placed where these results are discussed. A site map would be helpful in the study area section, but perhaps the map of Korea showing where Yeonchen is located would be sufficient.  
  • Further information on climate, reference vegetation patterns, topography, and dominant weather patterns could be helpful to fill out the study area section. These will also help the reader interpret questions about the combined impacts of topography and distance from pollution source.  
  • While the study area considered by the broader geospatial/landscape analyses appears to be evident in Figure 1, it is unclear how the vegetation survey plots were distributed in this space.  
  • The authors describe a “hypothetical belt transect installed by reflecting changes of the distance and the topographic conditions from the pollution source.” However, a figure depicting the plot layout within this hypothetical belt transect would be helpful for readers to interpret exactly what is meant by this.  
  • The authors interpret vegetation community patterns as a function of “degree of air pollution estimated by distances from the pollution source and topographic conditions. Are air pollution or atmospheric deposition records available that support this estimation framework?  
  • As noted above, the authors provide detailed description of vegetation survey plots and techniques, but it is still unclear how these plots are distributed across the landscape, and whether that sampling is reasonably representative.  
  • Late-successional reference forest is defined as oak-dominated, but unclear why the pre-industrial forests were pine dominated? This would be relevant to discuss in the introduction or in the study area description.  
  • How were the landscape ecological maps constructed and analyzed?  

Results:  

  • Because this study is concerned with 1971-2018, the information about the historical condition of the study area (I.e., pre-1970) seems best placed in the Study Area subsection of Methods and Materials.  
  • Figure 2 is helpful, but it is unclear how sampling plots would have been distributed across this transect. Also, were there multiple transects representative of various cardinal directions?  
  • I suggest revising the subsection headings in the results section to be consistent with the objectives listed in the introduction section.  
  • If I am interpreting figure 2 correctly, it looks like grassland --> shrubland --> grassland --> shrubland --> forest --> cropland --> forest --> shrubland --> grassland. It seems like oversimplification to interpret this as a trend from grassland to shrubland to forest with increased distance from the pollution source, especially given that the depicted sequence ends with grassland. (As noted later, suggest moving this figure to the Discussion—it would be more useful there as a somewhat simplified synthesis of all the data discussed.)  
  • Figure 3 is difficult to interpret. Axis 1 and 2 should include collections of variables, correct? Information about the environmental variables clustered in these axes would help with interpretation of this analysis. It is unclear what this analysis is contributing to the paper.  
  • Figure 5 is somewhat difficult to interpret because the legend linking species to their bar colors is somewhat too small to see the patterns. I suggest increasing the size of the legend to clarify this. It also appears that different pattern schemes are used in the second part of figure 5 than the first part--I suggest maintaining a consistent association of patterns with particular species to limit confusion. It may also help to restructure this figure to showcase the particular species of interest separately, rather than showing everything stacked together in the stacked-bar type. This alternate structure might also lend itself well to the way these data are actually being interpreted in the discussion section, which seems to track individual species.  
  • Figure 6: I think the arrow indicating the year the facility began operations is missing?  
  • Table 1: This table seems to repeat the data available in Figure 1. I suggest making this table supplemental and sticking with Figure 1, which is more impactful and easier to interpret.  

Discussion:  

  • Unclear whether the results of the DCA analysis are interpreted? Those results require further interpretation, but should be excluded or moved to supplemental material if they aren’t going to be discussed further.  
  • Figure 2 seems better placed here in the Discussion section, if it is a theoretical “typical” transect synthesizing data from across the multiple analyses discussed.  
  • Overall, the discussion section could be more explicitly connected to the data, and more detailed.  
  • This is especially true for the Restoration section at the end—this is some of the first conversation about restoration in the paper. It should be developed more fully and more explicitly connected to the data.  

Author Response

Response to reviewer’s comments

Reviewer 1 

Overall comments: The paper reports on shifts in vegetation over time as a result of air pollution sourced in industrial activity in the Yeocheon industrial complex in Korea. The research presents interesting and important findings and is certainly appropriate for Forests, with major revision recommended. The paper could benefit from some editing for grammar and usage in some places—syntax is at times awkward.  

☞ We revised our manuscript through English review.

 Specific comments:  

Abstract:  

  • The abstract is overly focused on the important results of the study, and lacks a treatment of the context and methods of the study. I suggest rewriting the abstract as a brief summary of the entire paper, including all major sections.  
  • ☞ We revised it by accepting reviewer’s comment. Lines 20 – 23.

Introduction:  

  • Pg 2, paragraph 3: “The study... have usually focused” (example of English usage problem: the subject and verb should agree in number)  
  • ☞ We revised it by accepting reviewer’s comment.
  • The authors describe some of the specific air pollution types and effects in the Materials and Methods section, but I suggest that some of this may be more appropriately relocated to the Introduction—for example, much of the paragraph on SO2 emission and its effect on soils. Also, is SO2 the only pollutant of concern in this region? This is the only specific pollutant mentioned.  In addition, we added other pollutant of concern in this area. Lines 145-146.
  • Methods and Materials:  
  • ☞ We addressed vegetation damage due to air pollution in this study area in Introduction section. We addressed this part in a perspective of ‘Site description’. So, we’d like to keep this content in ‘Study area’ subsection.
  • Figure 1 seems to be also the results of the analysis of landscape change, which is listed as the final objective in the introduction. This figure may be better placed where these results are discussed. A site map would be helpful in the study area section, but perhaps the map of Korea showing where Yeonchen is located would be sufficient.   
  • ☞ We moved Figure 1 by changing the Figure number (Figure 7) to ‘Results’ section by accepting reviewer’s comment.
  • Further information on climate, reference vegetation patterns, topography, and dominant weather patterns could be helpful to fill out the study area section. These will also help the reader interpret questions about the combined impacts of topography and distance from pollution source.   
  • ☞ We revised our manuscript by accepting reviewer’s comment. Lines 104 - 126.
  • While the study area considered by the broader geospatial/landscape analyses appears to be evident in Figure 1, it is unclear how the vegetation survey plots were distributed in this space.  
  • ☞ We expressed the location of the vegetation survey plots on a map (Figure 1) by accepting reviewer’s comment.
  • The authors describe a “hypothetical belt transect installed by reflecting changes of the distance and the topographic conditions from the pollution source.” However, a figure depicting the plot layout within this hypothetical belt transect would be helpful for readers to interpret exactly what is meant by this.
  • ☞ We revised our manuscript by expressing the location of the hypothetical belt transect on a map (Figure 1).
  • The authors interpret vegetation community patterns as a function of “degree of air pollution estimated by distances from the pollution source and topographic conditions.” Are air pollution or atmospheric deposition records available that support this estimation framework?  
  • ☞ We have just past data. However, pollution of this area in these days was significantly lower than that in the past, and no detailed measurement is made. So, we did such an interpretation based on the past data, which were published (Lee et al. 2004, An et al., 2015).
  • As noted above, the authors provide detailed description of vegetation survey plots and techniques, but it is still unclear how these plots are distributed across the landscape, and whether that sampling is reasonably representative.  
  • ☞ We prepared a map (Figure 1) including the locations of the vegetation survey plots and the belt transect additively.
  • Late-successional reference forest is defined as oak-dominated, but unclear why the pre-industrial forests were pine dominated? This would be relevant to discuss in the introduction or in the study area description.
  • ☞ This region is attributed to the temperate deciduous forest zone. So, we defined oak-dominated forest as the late successional reference forest instead pine dominated forest, which dominated forest of this area before construction of industrial complex. We added the explanation for the vegetation of this area in the ‘study area’ section. Lines 104 – 126.
  • How were the landscape ecological maps constructed and analyzed?  Results:  
  • ☞ We constructed the landscape ecological map based on aerial photo interpretation and field check. We explained the method in lines 167 to 169.
  • Because this study is concerned with 1971-2018, the information about the historical condition of the study area (I.e., pre-1970) seems best placed in the Study Area subsection of Methods and Materials.  
  • ☞ We addressed the information in ‘Study area’ section. Lines 104 – 126.
  • Figure 2 is helpful, but it is unclear how sampling plots would have been distributed across this transect. Also, were there multiple transects representative of various cardinal directions?  
  • ☞ We established two hypothetical belt transects as shown in Figure 1. In nature, variation is very severe, so we simplified it and expressed the trend as a diagrammatic model.
  • I suggest revising the subsection headings in the results section to be consistent with the objectives listed in the introduction section. 
  • ☞ We revised our manuscript by accepting reviewer’s comment.
  • If I am interpreting figure 2 correctly, it looks like grassland --> shrubland --> grassland --> shrubland --> forest --> cropland --> forest --> shrubland --> grassland. It seems like oversimplification to interpret this as a trend from grassland to shrubland to forest with increased distance from the pollution source, especially given that the depicted sequence ends with grassland. (As noted later, suggest moving this figure to the Discussion—it would be more useful there as a somewhat simplified synthesis of all the data discussed.)  
  • ☞ We revised our manuscript by accepting reviewer’s comment.
  • Figure 3 is difficult to interpret. Axis 1 and 2 should include collections of variables, correct? Information about the environmental variables clustered in these axes would help with interpretation of this analysis. It is unclear what this analysis is contributing to the paper.   
  • ☞ As we explain in the text, vegetation types established in the inland area are arranged in the upper part on the diagonal line that Axis â… and Axis â…¡ form, whereas vegetation types established in the coastal area in the lower parts of the line. On the other hand, vegetation types were changed from grassland through pine forest to the late successional oak forests in the upper part of the line, while from grassland through shrub land to the C. sinensis forest as move from the lower left to the upper right directions on the plane that Axis â… and Axis â…¡ form. From these results, we can interpret that Axis â…  means successional stage and Axis â…¡ reflects the distance from the coastal area. Lines 458 – 469.
  • Figure 5 is somewhat difficult to interpret because the legend linking species to their bar colors is somewhat too small to see the patterns. I suggest increasing the size of the legend to clarify this. It also appears that different pattern schemes are used in the second part of figure 5 than the first part--I suggest maintaining a consistent association of patterns with particular species to limit confusion. It may also help to restructure this figure to showcase the particular species of interest separately, rather than showing everything stacked together in the stacked-bar type. This alternate structure might also lend itself well to the way these data are actually being interpreted in the discussion section, which seems to track individual species. 
  • ☞ We revised Figure 5.
  •  
  • Figure 6: I think the arrow indicating the year the facility began operations is missing?  
  • ☞ We revised Figure 6.
  • Table 1: This table seems to repeat the data available in Figure 1. I suggest making this table supplemental and sticking with Figure 1, which is more impactful and easier to interpret.  
  • ☞ We moved Table 1 as the supplementary data (Appendix 1) by accepting reviewer’s comment.
  •  
  • Discussion:  
  • Unclear whether the results of the DCA analysis are interpreted? Those results require further interpretation, but should be excluded or moved to supplemental material if they aren’t going to be discussed further. 
  •  â˜ž We discussed the result of DCA ordination in ‘Discussion’ section by accepting reviewer’s comment. Lines 458 – 469.
  • Figure 2 seems better placed here in the Discussion section, if it is a theoretical “typical” transect synthesizing data from across the multiple analyses discussed.  
  • ☞ We moved Figure 2 to ‘Discussion’ section by accepting reviewer’s comment. Lines 446 – 457.
  • Overall, the discussion section could be more explicitly connected to the data, and more detailed.  
  • ☞ We revised ‘Discussion’ section by accepting reviewer’s comment. Lines 446 – 469. In addition, we added ‘Conclusion’ section to aid understanding of the readers..
  • This is especially true for the Restoration section at the end—this is some of the first conversation about restoration in the paper. It should be developed more fully and more explicitly connected to the data.  
  • ☞ We interpreted the vegetation change, which occur in this area in these days, as a passive restoration and evaluated that the speed is very slow compared with the active restoration.

Reviewer 2 Report

The manuscript assess vegetation changes that have occurred in the last decades around an industrial complex. The main argument is that pollution have sheer consequences on vegetation and it has the potential to alter vegetation composition at various spatial scales, up to the landscape. Vegetation changes are described in two stages. First stage comprise the period following establishment of the industrial complex, it is called vegetation decline. Second stage is considered when the vegetation start to recover and is called passive restoration. However, there is no clear information on the event that distinguish both stages. The authors explain that in the early 90’s pollutant concentration decrease until now, but no clear reason is stated. In addition, restoration, even passive, can only occur after the effect of the disturbance have ceased. It seems that the industrial complex is still growing. Moreover, it seems that socio-cultural changes have transformed the ways that local communities used the landscape, in particular the forest. It is therefore, not clear, how this pressure release could have confounded the effect of pollution.

The work is a complete assessment of the changes that have experience the vegetation. One strength is the different approaches that the authors use. However, most times the manuscript reads as a mixed of all these approaches and there is no clear line of argument among them.

Methodological section lacks information on the criteria of selection of plots in the different vegetation types. In addition, it is not clear, if different plots were sampled at different time steps or if they were the same. This information is crucial as the differences observed in vegetation in time may be due the different locations sampled. Moreover, there is no information on the date when the benchmark was sampled. Similarly, the number of trees cored is not given or the criteria of tree selection.

The statistical analysis should be improved. Only the DCA is described. Other questions that can be addressed are related to the patterns of biodiversity that are only analysed with a rank abundance curves. A transition matrix of landscape vegetation changes can also be added. Or the temporal trends of vegetation change. Even the age frequencies distribution among vegetation types can be compared.

Author Response

Response to reviewer’s comments

The manuscript assess vegetation changes that have occurred in the last decades around an industrial complex. The main argument is that pollution have sheer consequences on vegetation and it has the potential to alter vegetation composition at various spatial scales, up to the landscape. Vegetation changes are described in two stages. First stage comprise the period following establishment of the industrial complex, it is called vegetation decline. Second stage is considered when the vegetation start to recover and is called passive restoration. However, there is no clear information on the event that distinguish both stages. The authors explain that in the early 90’s pollutant concentration decrease until now, but no clear reason is stated. In addition, restoration, even passive, can only occur after the effect of the disturbance have ceased. It seems that the industrial complex is still growing. Moreover, it seems that socio-cultural changes have transformed the ways that local communities used the landscape, in particular the forest. It is therefore, not clear, how this pressure release could have confounded the effect of pollution.

 â˜ž Our result on landscape structure and change (Figure 7) shows that grassland began to appear as approach toward the industrial complex of the northern part in the late 1970s. And then, the area occupied by the grasslands increased continuously through the 1990s to the early 2000s. In addition, in 1991 shrub-land also appeared newly and the area increased to the early 2000s as in the case of grassland. We interpreted these degradation of forest as vegetation decline by synthesizing various references including papers that the corresponding author studied in this area.

Figure 7 also shows the vegetation changes from grassland and shrub-land to the forest. Figure 4 shows that the woody plants existed as the undergrowth in grasslands grew to shrub or tree levels. Those changes mean that vegetation quality was improved and thus we interpreted the results as a passive restoration due to the improvement of the environmental condition from the socio-economic changes in this area as Environmental Kuznets curve hypothesis suggests. In fact, the result obtained from the national observatory network for air pollution shows that air pollution is improved continuously in these days even though the scale of the industrial complex has increased.

On the other hand, this region is attribute to the temperate deciduous forest zone. Therefore, coniferous forest, which had established in this area is a kind of artificial product. In the past, residents of the area have constantly interfered with forests to get firewood, organic fertilizers, food for livestock, and farming instrument from them. The coniferous forest was established as a result of such artificial interference. Nowadays, however, human interference on forests has been greatly reduced as fuel has been replaced by fossil fuels, and other materials have also been replaced by the industrial products. The changes of coniferous forests to mixed or deciduous broadleaved forests are the result of these socioeconomic changes. We discussed the landscape change by considering the facts synthetically.

Lines 376 – 410, Appendix 1.

The work is a complete assessment of the changes that have experience the vegetation. One strength is the different approaches that the authors use. However, most times the manuscript reads as a mixed of all these approaches and there is no clear line of argument among them.

 â˜ž We explained changes of the distribution and composition of vegetation depending on the spatiotemporal changes of environmental condition by applying the ordination method. The spatial distribution and species composition of vegetation were usually dominated by the distance from the source of pollution (Figures 2 and 8).

Vegetation decline occurred in the past was explained based on the results of analyses on the annual ring of pine, which has easily discernible annual ring and the age structure of two dominant species (Figure 5). On the other hand, the vegetation change to be occurred in the future, a passive restoration, was explained based on the result of analysis on the size class distribution of dominant woody plants (Figure 4).

Furthermore, the trend is also confirmed by the analysis of landscape changes (Figure 7, Appendix 1).

Methodological section lacks information on the criteria of selection of plots in the different vegetation types. In addition, it is not clear, if different plots were sampled at different time steps or if they were the same. This information is crucial as the differences observed in vegetation in time may be due the different locations sampled. Moreover, there is no information on the date when the benchmark was sampled. Similarly, the number of trees cored is not given or the criteria of tree selection.

 â˜ž We showed the locations of the plots and the hypothetical belt transects installed for vegetation survey in Figure 1. We usually the same plots in 1991 and 2018 but some plots changed a little because the plots were damaged. So, we excluded a difference from the different locations. Information on the reference vegetation was collected in 2018 from in the stable forests, which reached to the late successional stage. We referred that the number of cores sampled was 10 and selected pine tree, which has easily discernible annual ring. Lines 157 – 158, 230.

The statistical analysis should be improved. Only the DCA is described.

☞ We added the result of statistical analysis on the difference of annual ring growth between the suppressed period and the other period. Lines 236 – 239, 322 – 323, Table 1.

Other questions that can be addressed are related to the patterns of biodiversity that are only analysed with a rank abundance curves.

☞ We added Shannon index for species diversity. Line 280. Figure 3.

A transition matrix of landscape vegetation changes can also be added. Or the temporal trends of vegetation change.

☞ Transition matrix of landscape change was expressed in Appendix 1.

Even the age frequencies distribution among vegetation types can be compared.

☞ We compared the size class frequency distribution (Figure 4).

Reviewer 3 Report

The paper addresses the effect of pollution on vegetation patterns in southern Korea. The topic is interesting and within the scope of Forest journal. However, the main conclusions are speculative which hamper the validity of any conclusions presented in the manuscript.

My first concern is the whole study is based on presumption that reported vegetation dynamics patterns are solely due to pollution, which is hardly convincing. The results presented do not back such claims, excepted some overlaps between vegetation patterns and pollution source locations, which is not enough to attribute vegetation changes to pollution. Many environmental factors as temperature and rainfall, play a crucial role in vegetation dynamics including growth mortality, succession and so one. Therefore, it seems to be surprising none of these variables is included in the study. Failing to do so, we obviously miss a careful assessment of variation due to climate and those presumably linked to pollution.

As a second concern, the paper is poorly written and needs a thorough edit for flow, consistency and language. Also, there are issues with Figure numbering.

  1. Introduction

“Air pollution that concentrations of substances of substances in the air”, weird sentence, please rephrase.

“Some woody plants are invading to grassland … late successional stage correspond on the symptoms”. Not understandable. Please rephrase.

  1. Materials and Methods

2.1. Study area

“(refer to Figure 7)”. Is it a mistake, the Fig you are referring to?

Two times Fig. 1 in Page 3 & Page 8.

2.2. Vegetation survey and data processing

“Field survey was practiced from” Please rephrase.

  1. Results

3.1. Landscape structure

“Landscape structure Forest of this region before construction … is mixed partially depending on site”. I don’t understand this sentence. Need to be rephrased.

“By the way, a mixed forest that pine and broad-leaved trees…”. Please rephrase.

I don’t know the point of all this section. In any way, we clearly see natural dynamics of forests that presumably overlaps to lower extent with change due to pollution.

3.2. Spatial distribution of vegetation

”Such a distributional trend … on Gwangyang bay”. There is no clear evidence backing this claim. In fact, all this section trying to link the pollution to the distribution of vegetation is speculative.

3.3. Vegetation sequence

Patterns of vegetation sequence as presented do no necessary derive from the effect of pollution.

3.4. Species diversity

I don’t see the Fig. 4.

3.6. Yearly changes of annual rings

“pine trees, which survived from air pollution damage” Speculative.

In addition, tree rings are sensitive to climate and I’m wondering why climate is not included in the study. By related climate and tree rings, it will be possible to assess changes in the patterns due to climate and changes not related it, in such a case we may related it to pollution.

3.7. Age distribution of two dominant species

“… most individuals consisting these two communities … natural regeneration of the damaged forest.” This is far fetch.

3.8. Landscape change

Natural forest dynamics are well known to drive the transition between broadleaves, mixed, and coniferous forests. This seems to me what we observe here. You are trying to justify all vegetation dynamics by pollution, which is hardly convincing.

  1. Discussion

The discussion reflects the results which are overall quite speculative.

Author Response

Response to reviewer’s comments

The paper addresses the effect of pollution on vegetation patterns in southern Korea. The topic is interesting and within the scope of Forest journal. However, the main conclusions are speculative which hamper the validity of any conclusions presented in the manuscript.

My first concern is the whole study is based on presumption that reported vegetation dynamics patterns are solely due to pollution, which is hardly convincing. The results presented do not back such claims, excepted some overlaps between vegetation patterns and pollution source locations, which is not enough to attribute vegetation changes to pollution. Many environmental factors as temperature and rainfall, play a crucial role in vegetation dynamics including growth mortality, succession and so one. Therefore, it seems to be surprising none of these variables is included in the study. Failing to do so, we obviously miss a careful assessment of variation due to climate and those presumably linked to pollution.

As a second concern, the paper is poorly written and needs a thorough edit for flow, consistency and language. Also, there are issues with Figure numbering.

☞ We revised our manuscript by adding climate data and relocating figures. Lines 102 – 124. We added Figure 1 and moved Figures 1 and 2 to Figures 7 and 8. We are waiting English review. Sooner or later, we will revise our manuscript based on the result.

  1. Introduction

“Air pollution that concentrations of substances of substances in the air”, weird sentence, please rephrase.

“Some woody plants are invading to grassland … late successional stage correspond on the symptoms”. Not understandable. Please rephrase.

☞ We revised our manuscript by accepting reviewer’s comments. Lines 53, 88 – 89.

  1. Materials and Methods“(refer to Figure 7)”. Is it a mistake, the Fig you are referring to?☞ We revised our manuscript by accepting reviewer’s comments.“Field survey was practiced from” Please rephrase. 
  2. ☞ We revised our manuscript by accepting reviewer’s comments. Line 166.
  3. 2.2. Vegetation survey and data processing
  4. Two times Fig. 1 in Page 3 & Page 8.
  5. 2.1. Study area
  1. Results“Landscape structure Forest of this region before construction … is mixed partially depending on site”. I don’t understand this sentence. Need to be rephrased.I don’t know the point of all this section. In any way, we clearly see natural dynamics of forests that presumably overlaps to lower extent with change due to pollution.3.2. Spatial distribution of vegetation☞ This part was written based on the results obtained from two hypothetical belt transects shown in Figure 1. The former paper that the corresponding author published addresses the pollution state of this area. This is the follow-up study of the paper.3.3. Vegetation sequence☞ In ‘Results’ section, we read the result and we interpreted that in relation to pollution of this area in ‘Discussion’ section.I don’t see the Fig. 4.3.6. Yearly changes of annual ringsIn addition, tree rings are sensitive to climate and I’m wondering why climate is not included in the study. By related climate and tree rings, it will be possible to assess changes in the patterns due to climate and changes not related it, in such a case we may related it to pollution.3.7. Age distribution of two dominant species☞ Of course, it can be understood so. But the period when most young individuals were recruited in both S. japonicus and P. densiflora communities corresponded correctly to the period when annual ring growth of the pine trees survived from air pollution was suppressed. Moreover, S. japonicus seldom forms a pure stand like in this area. In case of P. densiflora, natural regeneration of the forest rarely occurs unless large open spaces are provided. We made such an interpretation considered the information synthetically. Lines 316 – 336, Figures 5 and 6.Natural forest dynamics are well known to drive the transition between broadleaves, mixed, and coniferous forests. This seems to me what we observe here. You are trying to justify all vegetation dynamics by pollution, which is hardly convincing.In addition, we also interpreted a change from the coniferous forest to the mixed or deciduous ones as the successional change due to socio-economic changes. Lines 376 – 410, Figure 7, Appendix 1.
  2. ☞ As you know, a retrogressive succession from forest to grassland or shrub-land does not occur without any disturbance. But we couldn’t find any disturbance factor except pollution occurred after construction of the industrial complex in this area. Moreover, as was mentioned above, the air pollution state was very severe in this area.
  3. 3.8. Landscape change
  4. “… most individuals consisting these two communities … natural regeneration of the damaged forest.” This is far fetch.
  5. ☞ You are right. Tree ring growth is closely related to climate. But tree ring growth was suppressed for about 10 years without any special change in climate in this area. So, we interpreted the growth suppression as the result due to air pollution and in fact, air pollution in this area was very severe at that time. Many references cited in this manuscript explain the state. Figure 5. Table 1.
  6. “pine trees, which survived from air pollution damage” Speculative.
  7. ☞ The species rank – dominance curve shows not only species richness but also species evenness. So, we compared species diversities among vegetation types different in damage degree and between survey years based on the graph.
  8. 3.4. Species diversity
  9. Patterns of vegetation sequence as presented do no necessary derive from the effect of pollution.
  10.  
  11. ”Such a distributional trend … on Gwangyang bay”. There is no clear evidence backing this claim. In fact, all this section trying to link the pollution to the distribution of vegetation is speculative.
  12. ☞ We revised our manuscript by accepting reviewer’s comments. Lines 376 – 410.
  13. “By the way, a mixed forest that pine and broad-leaved trees…”. Please rephrase.
  14. 3.1. Landscape structure
  1. Discussion☞ We tried to revise “Discussion’ section as much as possible based on the results that we collected and the references related directly to this study. Lines 412 – 575. In addition, we added ‘Conclusion’ section to aid understanding of the readers. Lines 576 – 589.
  2. The discussion reflects the results which are overall quite speculative.

Round 2

Reviewer 1 Report

Review_ forests-795091_R1 

General comments:  

I think the paper is much improved; I better understand the authors’ organizational structure and main points, and I think the conclusions are well-evidenced in the data displayed, and reasonable. I suggest some additional editing for grammar and usage (specifics below). The only major concern I still have is the lack of replication in the sampling approach. I think the results are still meaningful and reasonable, but the limitations imposed by the sampling design should be addressed.  

Specific comments:  

Abstract:  

Line 24-27: Sentence beginning “The spatial distribution...” Rephrase—the spatial distribution is not appearing in an order; rather, the vegetation is appearing in an order. Suggested rephrasing: “Vegetative communities tended to be spatially distributed (with increasing distance from the pollution source) as follows: grassland, shrubland...”  

Line 29: sentence beginning “Species richness evaluated...” Edit to “species richness in forests was intermediate” to avoid subject/verb confusion  

Line 44: missing word “these results are line...”  

Introduction:  

Line 52: edit to “Air pollution, which occurs when....”  

Line 89: missing word? “...In the late successional stage correspond.”  

Materials and Methods:  

Line 107: some grammar/punctuation issues here. Suggested edit: “...run in the northeast direction, and a third ridge, which stretches from Mt. Cheonseong through Mt. Bonghwa to Mt. Horang, runs in the northwest direction, connecting the other two.”  

Line 127: Unclear wording: “Yeocheon industrial complex is the representative heavy-chemical industrial complexes of Korea...” Rephrase to clarify whether Yeochen is representative of” this group of heavy-chemical industrial complexes, or is “the representative industrial complex.”  

Line 127: clarify whether the “standard value” for SO2 levels is the target ambient SO2 concentration set by regulation, or the “typical” or “usual concentration 

Lines 143-146: condense to previous paragraph.  

Lines 178, 182: Figure 1 depicts only five total plots, but the text indicates multiple plots per vegetation class. Revise language to clarify—e.g., subplots rather than plots if talking about sampling quadrats within the given stands. Also, if I am interpreting this experimental design correctly, there was no replication—there were not multiple sampled transects that represented the whole range of surveyed plant communities. Address the lack of replication intentionally in text, and discuss the implications for interpretation and application of results.  

Lines 241-245: Redundant—revise for conciseness  

Results:  

Figure 2: It is still unclear what variables are sorted in the different axes of the DCA. The authors identify slope position as a variable—is that the only variable involved in these axes? Consider including in the figure caption an indication of which variables are included in the axes to aid the reader in interpreting the figure.  

Figure 3: What are the numbers in the box inset in the figure? The inset is not mentioned in the figure caption, but I’m guessing it is a table of Shannon indices?  

Table 1: Indicate the time ranges considered suppressed and non-suppressed. Also, include units in the “mean” column  

Line 336: Typo: “rdamaged”  

Line 396: Typo: “area of occupied...”  

Lines 458: The further interpretation of the DCA is very helpful; however, I still suggest including some interpretative details about the variables included in the axes in the figure caption itself.  

Author Response

Response to reviewer’s comments

Reviewer 1 

I think the paper is much improved; I better understand the authors’ organizational structure and main points, and I think the conclusions are well-evidenced in the data displayed, and reasonable. I suggest some additional editing for grammar and usage (specifics below). The only major concern I still have is the lack of replication in the sampling approach. I think the results are still meaningful and reasonable, but the limitations imposed by the sampling design should be addressed.  

Specific comments:  

Abstract:  

Line 24-27: Sentence beginning “The spatial distribution...” Rephrase—the spatial distribution is not appearing in an order; rather, the vegetation is appearing in an order. Suggested rephrasing: “Vegetative communities tended to be spatially distributed (with increasing distance from the pollution source) as follows: grassland, shrubland...”  

☞ We revised this part by accepting reviewer’s comment. Lines 24 – 27.

Line 29: sentence beginning “Species richness evaluated...” Edit to “species richness in forests was intermediate” to avoid subject/verb confusion  

☞ We revised this part by accepting reviewer’s comment. Line 30.

Line 44: missing word “these results are line...”  

☞ We revised this part by accepting reviewer’s comment. Line 46.

Introduction:  

Line 52: edit to “Air pollution, which occurs when....”  

☞ We revised this part by accepting reviewer’s comment. Line 54.

Line 89: missing word? “...In the late successional stage correspond.”  

☞ We revised this part so that the readers would not misunderstood. Line 91.

Materials and Methods:  

Line 107: some grammar/punctuation issues here. Suggested edit: “...run in the northeast direction, and a third ridge, which stretches from Mt. Cheonseong through Mt. Bonghwa to Mt. Horang, runs in the northwest direction, connecting the other two.”  

☞ We revised this part by accepting reviewer’s comment. Lines 107 - 110.

Line 127: Unclear wording: “Yeocheon industrial complex is the representative heavy-chemical industrial complexes of Korea...” Rephrase to clarify whether Yeochen is “representative of” this group of heavy-chemical industrial complexes, or is “the representative industrial complex.”  

☞ The former is right. That is the representative heavy-chemical industrial complexes.

Line 127: clarify whether the “standard value” for SO2 levels is the target ambient SO2 concentration set by regulation, or the “typical” or “usual concentration.  

☞ ”Standard value” means the target ambient SO2 concentration set by regulation. So, we added the explanation. Line 146.

Lines 143-146: condense to previous paragraph.  

☞ We revised this part by accepting reviewer’s comment. Lines 137 - 147.

Lines 178, 182: Figure 1 depicts only five total plots, but the text indicates multiple plots per vegetation class. Revise language to clarify—e.g., subplots rather than plots if talking about sampling quadrats within the given stands. Also, if I am interpreting this experimental design correctly, there was no replication—there were not multiple sampled transects that represented the whole range of surveyed plant communities. Address the lack of replication intentionally in text, and discuss the implications for interpretation and application of results.  

☞ We revised this part by accepting reviewer’s comment. Lines 183 – 199. We addressed the lack of replication and reflected that in discussion of the results. Lines 161 – 162 and 486 - 488.

Lines 241-245: Redundant—revise for conciseness  

☞ We revised this part by accepting reviewer’s comment. Lines 248 - 252.

Results:  

Figure 2: It is still unclear what variables are sorted in the different axes of the DCA. The authors identify slope position as a variable—is that the only variable involved in these axes? Consider including in the figure caption an indication of which variables are included in the axes to aid the reader in interpreting the figure.  

☞ We revised the caption of Figure 2 by accepting reviewer’s comment.

Figure 3: What are the numbers in the box inset in the figure? The inset is not mentioned in the figure caption, but I’m guessing it is a table of Shannon indices?  

☞ Yes, you are right. So, we added the explanation in the figure caption.

Table 1: Indicate the time ranges considered suppressed and non-suppressed. Also, include units in the “mean” column  

☞ We revised this part by accepting reviewer’s comment.

Line 336: Typo: “rdamaged”  

☞ We revised this part by accepting reviewer’s comment. Line 348.

Line 396: Typo: “area of occupied...”  

☞ We revised this part by accepting reviewer’s comment. Line 408.

Lines 458: The further interpretation of the DCA is very helpful; however, I still suggest including some interpretative details about the variables included in the axes in the figure caption itself.  

☞ We revised this part by accepting reviewer’s comment. Lines 481 – 485. Caption of Figure 2.

Reviewer 2 Report

The authors have improved the manuscript. I only have minor comments.

L44, add “in” before line

L138, this is already said in line 128

L146, better to add a short explanation, rather than only include the reference

L573, please include de method. This will help readers understand your conclusions without previous experience on the topic.

Author Response

Response to reviewer’s comments

Reviewer 2

The authors have improved the manuscript. I only have minor comments.

L44, add “in” before line

☞ We revised this part by accepting reviewer’s comment. Line 46.

L138, this is already said in line 128

☞ We revised our manuscript to avoid duplication. Lines 130 – 147.

L146, better to add a short explanation, rather than only include the reference

☞ We revised this part by accepting reviewer’s comment. Lines 148 - 151.

L573, please include de method. This will help readers understand your conclusions without previous experience on the topic.

☞ We revised this part by accepting reviewer’s comment. Lines 598 - 601.

Reviewer 3 Report

The paper has improved and the concerns earlier raised have been properly addressed. The lake of replications for some measurements may be briefly discussed in the light of the validity of the result obtained. In addition, the paper needs a proofreading for English language. Please find some specific comments below.

L88-89 “For instance, grassland … ”. This sentence seems weird, is there something missing?

L125-126

1) I understood later in the paper that oak forest has been picked up as a reference forest for species composition and diversity assessment. Please clarify.

2) “damaged vegetation types”, do you mean the vegetation close to the source of pollution?

3) Is C. sinensis also a reference forest? If yes, you may indicate for which variable because it Is not clear in the results.

L148-162 There are no replications for some measurements. This may briefly be discussed in the light of the validity of some results obtained.

L260-261“…whereas stands of P. thunbergii and P. densiflora little changed…”. Compared to what?

Author Response

Response to reviewer’s comments

Reviewer 3

The paper has improved and the concerns earlier raised have been properly addressed. The lake of replications for some measurements may be briefly discussed in the light of the validity of the result obtained. In addition, the paper needs a proofreading for English language. Please find some specific comments below.

L88-89 “For instance, grassland … ”. This sentence seems weird, is there something missing?

☞ We revised this part by accepting reviewer’s comment. Lines 90 – 91.

L125-126

1) I understood later in the paper that oak forest has been picked up as a reference forest for species composition and diversity assessment. Please clarify.

☞ We clarified it. Lines 122 – 128.

2) “damaged vegetation types”, do you mean the vegetation close to the source of pollution?

☞ Yes, you are right. We revised this part to avoid misunderstanding. Lines 128 -129.

3) Is C. sinensis also a reference forest? If yes, you may indicate for which variable because it Is not clear in the results.

☞ Yes, we regarded C. sinensis as another reference forest. We addressed that C. sinensis is a late successional species in this area and replace the damaged vegetation as the environmental condition are improved in these days in various parts in our manuscript. Lines 261 – 263, 266 – 269, 470 – 485, and ss0 – 552.

L148-162 There are no replications for some measurements. This may briefly be discussed in the light of the validity of some results obtained.

☞ We revised this part by accepting reviewer’s comment. Lines 179 – 188. We addressed the lack of replication and reflected that in discussion of the results. Lines 160 – 161 and 483 - 487.

L260-261“…whereas stands of P. thunbergii and P. densiflora little changed…”. Compared to what?

☞ This means that there was no remarkable change in species composition of two stands rather than compare to another stands.